# Genetic architecture of human plasma lipidome and its link to cardiovascular disease

Rubina Tabassum et al.[#]

Understanding genetic architecture of plasma lipidome could provide better insights into lipid metabolism and its link to cardiovascular diseases (CVDs). Here, we perform genome-wide association analyses of 141 lipid species (n = 2,181 individuals), followed by phenome-wide scans with 25 CVD related phenotypes (n = 511,700 individuals). We identify 35 lipid-species-associated loci (P < 5 ×10$^{-8}$), 10 of which associate with CVD risk including five new loci-*COL5A1*, *GLTPD2*, *SPTLC3*, *MBOAT7* and *GALNT16* (false discovery rate<0.05). We identify loci for lipid species that are shown to predict CVD e.g., *SPTLC3* for CER(d18:1/24:1). We show that lipoprotein lipase (LPL) may more efficiently hydrolyze medium length triacylglycerides (TAGs) than others. Polyunsaturated lipids have highest heritability and genetic correlations, suggesting considerable genetic regulation at fatty acids levels. We find low genetic correlations between traditional lipids and lipid species. Our results show that lipidomic profiles capture information beyond traditional lipids and identify genetic variants modifying lipid levels and risk of CVD.

---

Cardiovascular diseases (CVDs) encompass many pathological conditions of impaired heart function, vascular structure and circulatory system. CVDs are the leading cause of mortality and morbidity worldwide[1], necessitating the need for better preventive and predictive strategies. Plasma lipids, the well-established heritable risk factors for CVDs[2], are routinely monitored to assess CVD risk. Standard lipid profiling measures traditional lipids (referred to LDL-C, HDL-C, total triglycerides and total cholesterol), but does not capture the functionally and chemically diverse molecular components—the lipid species[3]. These molecular lipid species may independently and specifically affect different manifestations of CVD, such as ischaemic heart disease and stroke. Lipid species including cholesterol esters (CEs), lysophosphatidylcholines (LPCs), phosphatidylcholines (PCs), phosphatidylethanolamines (PEs), ceramides (CERs), sphingomyelins (SMs) and triacylglycerols (TAGs) potentially improve CVD risk assessment over traditional lipids[4–9].

Understanding of the genetic architecture and genetic regulation of these lipid species could help guide tool development for CVD risk prediction and treatment. Genetic studies of traditional lipids have identified over 250 genomic loci and improved our understanding of CVD pathophysiology[10,11]. For the majority of the lipid loci, however, their effects on detailed lipidome beyond traditional lipids are unknown. Only a few studies have reported genetic associations for lipid species either through studies on subsets of the lipidome[12,13] or GWASs on metabolome[14–20].

In light of the limited information about the genetics of lipidomic profiles and their relationship with CVDs, we carried out a GWAS of lipidomic profiles of 2181 individuals using ~9.3 million genetic markers followed by PheWAS including 25 CVD-related phenotypes in up to 511,700 individuals (Fig. 1). We aimed to (1) determine heritability of lipid species and their genetic correlations; (2) identify genetic variants influencing the plasma levels of lipid species; (3) test the relationship between identified lipid–species-associated variants and CVD manifestations and (4) gain mechanistic insights into established lipid variants. We find that lipid species are heritable, suggesting a considerable role of endogenous regulation in lipid metabolism. We report association of new genomic loci with lipid species and CVD risk in humans. In addition to enhancing the current understanding of genetic regulation of circulating lipids, our study emphasises the need of lipidomic profiling in identifying additional variants influencing lipid metabolism.

## Results

**Heritability of lipid species**. First, we determined SNP-based heritability for each of the lipid species and traditional lipids using genetic relationship matrix for all the study participants. The demographic characteristics of the study participants are provided in Supplementary Table 1. SNP-based heritability estimates ranged from 0.10 to 0.54 (Fig. 2a; Supplementary Table 2), showing considerable variation across lipid classes (Fig. 2b), with similar trends as reported previously[21,22]. CERs showed the greatest estimated heritability (median = 0.39, range = 0.35–0.40), whereas phosphatidylinositols (PIs) showed the least heritability (median = 0.19, range = 0.11–0.31). Sphingolipids had higher heritability than glycerolipids ranging from 0.24 to 0.41 (Fig. 2b), which is similar to a previous study that reported higher heritability for sphingolipids ranging from 0.28 to 0.53 estimated based on pedigrees[21]. Lipids containing polyunsaturated fatty acids, particularly C20:4, C20:5 and C22:6, had significantly higher heritability compared with other lipid species (Fig. 2c). For instance, PC (17:0;0–20:4;0) and LPC (22:6;0) had the highest heritability (> 0.50), whereas PC (16:0;0–16:1;0) and PI (16:0;0–18:2;0) had the lowest heritability estimates (< 0.12) (Supplementary Table 2).

**Genetic correlations between lipid species**. Longer, polyunsaturated lipids (those with four or more double bonds) had stronger genetic correlations with each other than with other lipid species (Supplementary Fig. 1, Supplementary Data 1). This can be seen in the hierarchical clustering based on genetic correlations that segregate TAG subspecies into two clusters based on carbon content and degree of unsaturation (Fig. 2d). These patterns were not seen in phenotypic correlations that were estimated based on the plasma levels of lipid species (Supplementary Fig. 2).

We observed low phenotypic and genetic correlation between traditional lipids and molecular lipid species, except strong positive genetic correlations of triglycerides with TAGs and DAGs (average $r = 0.88$) (Fig. 3). However, triglycerides had low genetic correlation with other lipid species (average (abs) $r = 0.26$). HDL-C and LDL-C levels had low genetic and phenotypic

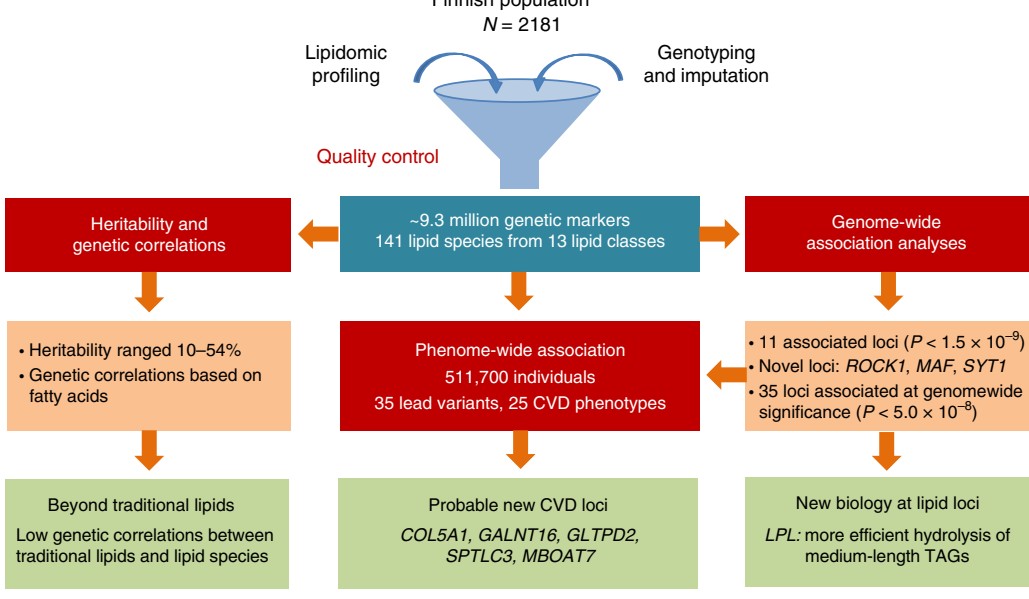

**Fig. 1** Study design and work flow. The figure illustrates the study design and key findings of the study

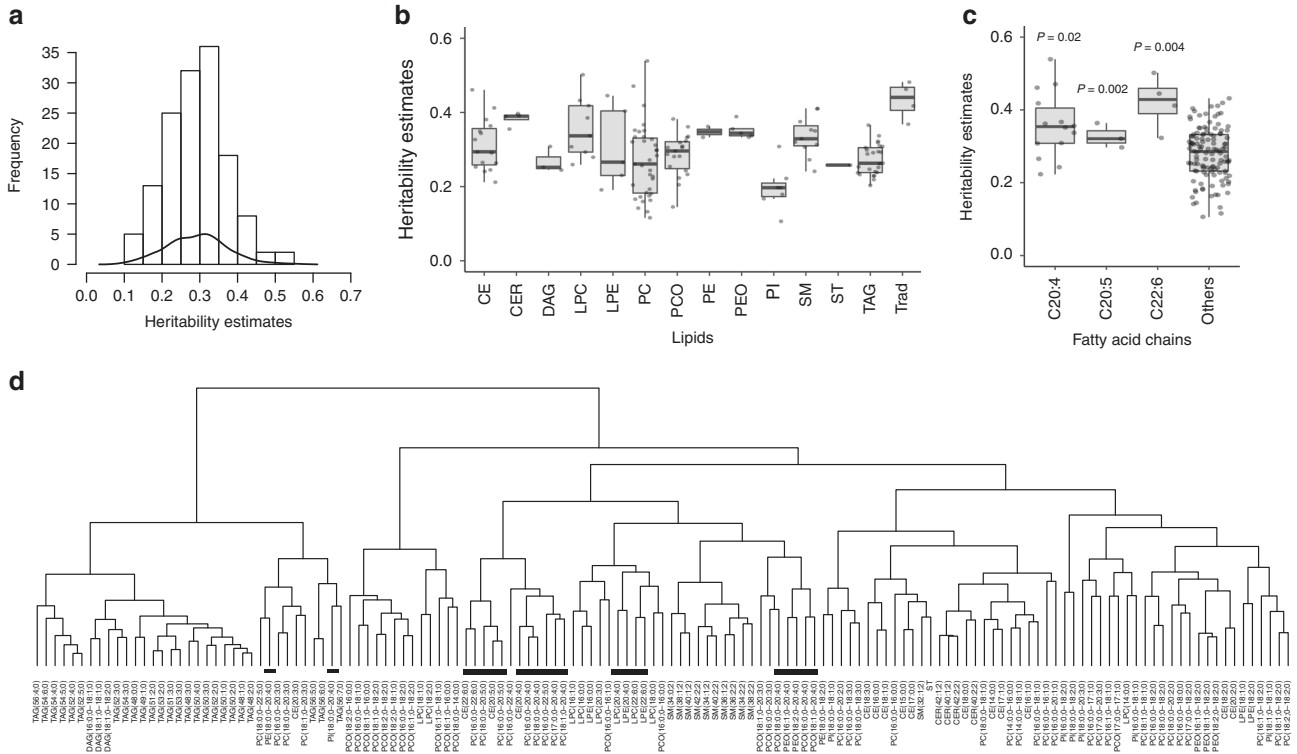

**Fig. 2** Heritability of lipidomic profiles and genetic correlations among the lipid species. **a** Histogram and kernel density curve showing the distribution of heritability estimates across all the lipid species. **b** Boxplot showing the heritability estimates in each lipid class. **c** Boxplot showing comparison of the median heritability estimates of lipid species containing C20:4, C20:5 and C22:6 acyl chains and all others. The P-values were calculated using the Wilcoxon rank-sum test. **d** Hierarchical clustering of lipid species based on genetic correlations among lipid species. Lipids containing polyunsaturated fatty acids C20:5, C20:4 and C22:6 are highlighted with black bars. The data presented in the boxplots represent the interquartile range (IQR) defined by the bounds of the box with the median (middle line of the box) and whiskers extending to the largest/smallest values no further than 1.5 times the IQR. CER ceramide, DAG diacylglyceride, LPC lysophosphatidylcholine, LPE lysophosphatidylethanolamine, PC phosphatidylcholine, PCO phosphatidylcholine-ether, PE phosphatidylethanolamine, PEO phosphatidylethanolamine-ether, PI phosphatidylinositol, CE cholesteryl ester, SM sphingomyelin, ST sterol, TAG triacyglycerol, Trad traditional lipids

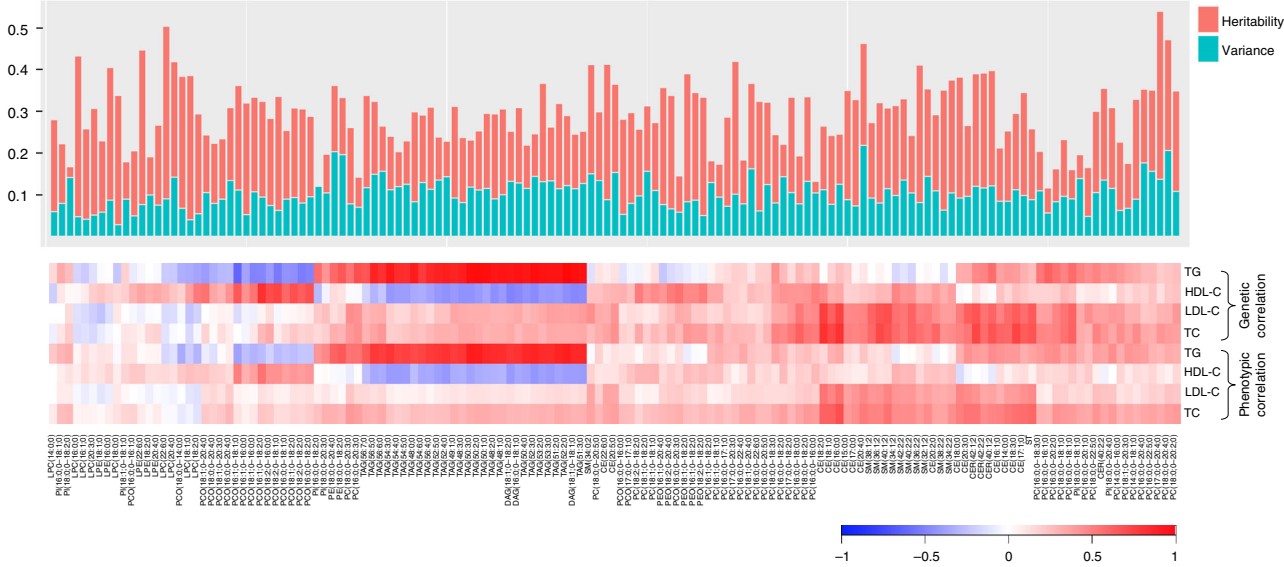

**Fig. 3** Lipidomic profiles capture information beyond traditional lipids. The genetic and phenotypic correlations between traditional lipids and molecular lipid species are shown in lower panel. The bar plot in the upper panel shows the heritability estimates of each lipid species (red bars) and the variance explained by all the known loci together (green bars). The lipid species are ordered based on the hierarchical clustering showing the correlations between the lipid species and traditional lipids. TC total cholesterol, TG triglycerides

**Table 1 Genomic loci associated with molecular lipid species at genome-wide significance**

| SNP | Position | Gene | Change | Ref | Alt | AF | Lipid species | Effect | SE | P |
|---|---|---|---|---|---|---|---|---|---|---|
| rs201385366 | 1:897866 | KLHL17 | Intronic | C | T | 0.019 | LPE(22:6;0) | −0.87 | 0.16 | $3.6 \times 10^{-8}$ |
| rs187163948 | 1:14399146 | KAZN* | Intronic | G | A | 0.011 | TAG(53:3;0) | 0.95 | 0.17 | $3.5 \times 10^{-8}$ |
| rs76866386# | 2:44075483 | ABCG5/8 | Intronic | T | C | 0.077 | CE(20:2;0) | −0.39 | 0.06 | $3.9 \times 10^{-10}$ |
| rs58029241 | 2:98701245 | VWA3B* | Intergenic | T | A | 0.062 | TAG(50:1;0) | 0.37 | 0.07 | $1.9 \times 10^{-8}$ |
| rs13070110 | 3:21393248 | ZNF385D* | Intergenic | T | C | 0.085 | Total CER | 0.33 | 0.06 | $3.9 \times 10^{-9}$ |
| rs10212439 | 3:142655053 | PAQR9 | Intergenic | T | C | 0.602 | PI(18:0;0-18:1;0) | 0.18 | 0.03 | $3.1 \times 10^{-8}$ |
| rs13151374 | 4:8122221 | ABLIM2* | Intronic | G | A | 0.153 | TAG(51:0;0) | 0.25 | 0.04 | $3.7 \times 10^{-8}$ |
| rs186689484 | 4:97033701 | PDHA2* | Intergenic | G | A | 0.051 | TAG(52:4;0) | −0.40 | 0.07 | $4.2 \times 10^{-8}$ |
| rs543895501 | 6:74120350 | DDX43* | Intronic | C | T | 0.013 | Total LPC | 0.87 | 0.16 | $2.9 \times 10^{-8}$ |
| rs4896307 | 6:138297840 | TNFAIP3* | Intergenic | C | T | 0.216 | PCO(16:1;0-16:0;0) | −0.23 | 0.04 | $3.3 \times 10^{-8}$ |
| rs534693155 | 7:101081274 | COL26A1* | Intronic | A | G | 0.010 | LPC(16:1;0) | 1.24 | 0.23 | $3.9 \times 10^{-8}$ |
| rs10281741 | 7:157793122 | PTPRN2* | Intronic | G | C | 0.225 | TAG(54:6;0) | 0.21 | 0.04 | $2.2 \times 10^{-8}$ |
| rs1478898 | 8:11395079 | BLK* | Intronic | G | A | 0.440 | PC(16:0;0-16:0;0) | 0.17 | 0.03 | $2.5 \times 10^{-8}$ |
| rs11570891 | 8:19822810 | LPL | Intronic | C | T | 0.075 | TAG(52:3;0) | −0.33 | 0.06 | $2.9 \times 10^{-8}$ |
| rs146717710 | 9:137549865 | COL5A1* | Intronic | C | T | 0.011 | PC(16:0;0-16:1;0) | −1.03 | 0.19 | $2.8 \times 10^{-8}$ |
| rs140645847 | 10:118863255 | SHTN1* | Intronic | G | T | 0.101 | LPE(20:4;0) | −0.32 | 0.06 | $3.3 \times 10^{-8}$ |
| rs28456# | 11:61589481 | FADS2 | Intronic | A | G | 0.405 | CE(20:4;0) | −0.59 | 0.03 | $1.1 \times 10^{-77}$ |
| rs964184 | 11:116648917 | APOA5 | Intergenic | G | C | 0.855 | TAG(52:3;0) | −0.258 | 0.045 | $9.5 \times 10^{-9}$ |
| rs10790495 | 11:122198706 | MIR100HG* | Intronic | A | G | 0.590 | TAG(56:4;0) | −0.20 | 0.04 | $2.1 \times 10^{-8}$ |
| rs117388573# | 12:78980665 | SYT1* | Intergenic | A | G | 0.020 | LPC(14:0;0) | −0.77 | 0.13 | $9.8 \times 10^{-10}$ |
| rs512948 | 13:52374489 | DHRS12* | Intronic | T | C | 0.225 | LPE(18:2;0) | −0.22 | 0.04 | $1.4 \times 10^{-8}$ |
| rs8008070# | 14:64233720 | SYNE2 | Intronic | A | T | 0.133 | SM(32:1;2) | 0.48 | 0.05 | $2.9 \times 10^{-26}$ |
| rs3902951 | 14:69789755 | GALNT16 | Intronic | T | G | 0.361 | PEO(18:1;0-18:2;0) | 0.19 | 0.03 | $1.9 \times 10^{-8}$ |
| rs35861938 | 15:45637343 | GATM* | Intergenic | T | C | 0.398 | PCO(18:2;0-18:1;0) | 0.18 | 0.03 | $2.7 \times 10^{-8}$ |
| rs261290# | 15:58678720 | LIPC | Intronic | T | C | 0.617 | PE(18:0;0-20:4;0) | −0.37 | 0.03 | $4.0 \times 10^{-31}$ |
| rs35221977# | 16:79563576 | MAF* | Intronic | G | C | 0.054 | LPC(16:0;0) | −0.46 | 0.08 | $1.3 \times 10^{-9}$ |
| rs79202680# | 17:4692640 | GLTPD2 | Intronic | G | T | 0.032 | SM(34:0;2) | −0.85 | 0.09 | $3.4 \times 10^{-22}$ |
| rs143203352 | 17:77293933 | RBFOX3* | Intronic | T | C | 0.024 | PC(16:0;0-18:1;0) | 0.60 | 0.11 | $3.2 \times 10^{-8}$ |
| rs151223356# | 18:18627427 | ROCK1* | Intronic | A | C | 0.013 | LPC(14:0;0) | 0.97 | 0.15 | $1.9 \times 10^{-10}$ |
| rs7246617# | 19:8272163 | CERS4 | Intergenic | G | A | 0.402 | SM(38:2;2) | 0.25 | 0.03 | $2.5 \times 10^{-15}$ |
| rs2455069 | 19:51728641 | CD33* | Missense | A | G | 0.383 | TAG(52:5;0) | −0.19 | 0.03 | $9.3 \times 10^{-9}$ |
| rs8736# | 19:54677189 | MBOAT7 | UTR | C | T | 0.388 | PI(18:0;0-20:4;0) | −0.38 | 0.03 | $9.8 \times 10^{-28}$ |
| rs4374298 | 19:55738746 | TMEM86B* | Synonymous | G | A | 0.166 | PEO(16:1;0-20:4;0) | −0.25 | 0.04 | $2.3 \times 10^{-8}$ |
| rs364585# | 20:12962718 | SPTLC3 | Intergenic | A | G | 0.670 | Total CER | −0.20 | 0.03 | $9.1 \times 10^{-10}$ |
| rs186680008 | 22:39754367 | SYNGR1* | Intronic | A | C | 0.015 | CE(20:3;0) | −0.81 | 0.15 | $2.6 \times 10^{-8}$ |

Ref reference allele, Alt alternate allele, AF alternate allele frequency, SE standard error, UTR untranslated region
The strongest association between SNP and lipid species in the genome-wide significant loci ($P < 5.0 \times 10^{-8}$) are presented. The P-values were calculated from the meta-analyses using the inverse variance weighted method for fixed effects. The study-wide significant associations are marked by hash symbol. The SNPs are annotated to the nearest gene if identified in this study (marked by asterisk symbol) or to previously known gene if in linkage disequilibrium with the known loci for any lipid measure. The effect sizes presented are change in standard deviation of the lipid species per alternate allele. Chromosomal positions are based on hg19 reference sequence

correlations with most of the lipid species (Fig. 3; Supplementary Data 1). Consistently, all of the known lipid variants explained 2–21% of variances in plasma levels of various lipid species, with the least variance accounting for LPCs (Fig. 3). To rule out the possibility that lipid-lowering medications resulted in the observed low genetic correlations between traditional lipids and lipid species, we also calculated the genetic correlations after excluding the individuals using lipid lowering medications ($N = 172$). This re-analysis provided the similar results as the primary analysis (Supplementary Fig. 3). It is to be noted that this sample size might not provide sufficient power for heritability estimations in unrelated samples. Our study also included the family samples which provides higher statistical power in heritability estimation than unrelated samples.

**Lipid species associated variants.** Next, we performed the genome-wide association analyses for 141 lipid species with ~9.3 million genetic markers. We identified 2817 associations between 518 variants located within 11 genomic loci (1MB blocks) and 42 lipid species from 10 lipid classes at study-wide significance ($P < 1.5 \times 10^{-9}$ accounting for 34 principal components that explain 90% of the variance in lipidome) (Table 1; Supplementary Data 2, 3). These included three new loci (ROCK1, MAF and SYT1) that are not previously reported for any lipid measure or related metabolite (Fig. 4). Among the new loci, the strongest association was at an intronic variant rs151223356 near ROCK1 with short acyl-chain LPC(14:0,0) ($P = 1.9 \times 10^{-10}$). ROCK1 encodes for a serine/threonine kinase that plays key role in glucose metabolism[23]. In line with our

observation of higher heritability for lipids with C20:4, C20:5 or C22:6 acyl chains, we detected associations for 15 out of 21 lipids with these acyl chains.

We also replicated the previous associations of FADS2, SYNE2, LIPC, CERS4 and MBOAT7 with the same lipid species[13–20]. The previously reported associations at the known loci identified in previous metabolomics GWASs are provided in Supplementary Data 4. This information was obtained from the databases-SNiPA (http://snipa.org) using block annotation and PhenoScanner v2 (http://www.phenoscanner.medschl.cam.ac.uk/), and were manually curated to include associations from literature search. In addition, we also identified new locus–lipid species associations at previously reported lipid loci including new associations of variants at ABCG5/8 with CE (20:2;0) ($P = 3.9 \times 10^{-10}$), MBOAT7 with PI (18:0;0–20:3;0) ($P = 3.0 \times 10^{-12}$) and GLTPD2 with SM (34:0;2) ($P = 3.4 \times 10^{-22}$) (Supplementary Data 2, 3).

Further, we systematically evaluated the associations of variants previously identified in metabolomics GWAS (126 variants from 46 loci available in our data set out of 132 reported) with 141 lipid species. Of these known variants, 76 variants from 12 loci showed association with 98 different lipid species with $P < 3.2 \times 10^{-5}$ (correcting for 46 loci and 34 PCs for lipid species) (Supplementary Data 5). Of the 134 previously reported variant–lipid species pair associations that could be examined in our data set, 94 of such associations were replicated with the same direction of effect with $P < 3.7 \times 10^{-4}$ (accounting for 134 comparisons) in our study (Supplementary Data 6).

In addition, 24 further loci were associated with at least one lipid species at regularly used genome-wide significance level ($1.5 \times 10^{-9} > P < 5.0 \times 10^{-8}$). Among these additional loci, 13 loci

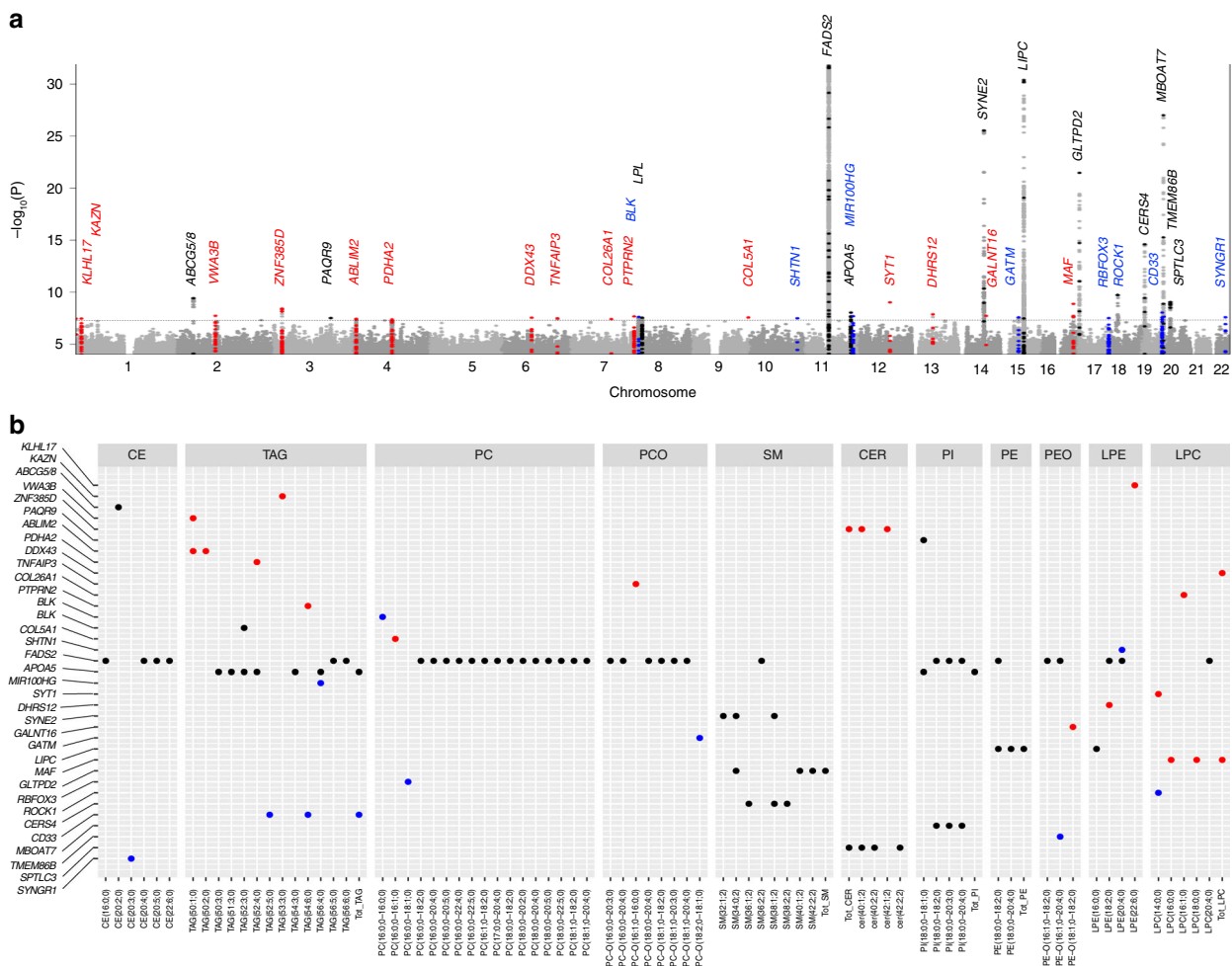

**Fig. 4** Genetic architecture of the lipidome. **a** Manhattan plot showing associations for all 141 lipid species. Only the associations with $P < 1.0 \times 10^{-4}$ in the meta-analysis and consistent in directions in all three batches are plotted. The y-axis is capped at $-\log_{10} P$-value $= 30$ for better representation of the data. The dotted line represents the threshold for genome-wide significant associations at $P < 5.0 \times 10^{-8}$. **b** Genome-wide significant associations between the identified lipid species-associated loci and lipid species showing effect of the loci on the lipidome. The plotted P-values were calculated from the meta-analyses using the inverse variance weighted method for fixed effects. New hits with $P < 5.0 \times 10^{-8}$ are shown as red dots, new independent hits in previously reported loci are presented as blue dots and hits in previously known loci are presented as black dots

were located in genomic regions not previously reported for any lipid measure or related metabolite, and 8 loci were located near known loci for lipids but were independent of any previously reported variant (Table 1; Supplementary Data 3). The regional association plots for all 35 loci with $P < 5.0 \times 10^{-8}$ are presented in Supplementary Data 7, and the genotype–phenotype relationships for the lead variants in these 35 loci are provided in Supplementary Fig. 4.

**Relationship between identified variants and risk of CVD.** As many of the lipid species have previously been shown to predict CVD risk, we determined if the variants associated with lipid species affect individuals' susceptibility to CVD-related phenotypes in FinnGen and UK Biobank cohorts. We identified 25 CVD-related phenotypes from the clinical outcomes derived from health registry data in the FinnGen and UK Biobanks (Supplementary Table 3). The follow-up PheWAS analyses included lead variants from all of the 35 independent loci that showed associations with $P < 5.0 \times 10^{-8}$ (Table 1). Overall, 10 of the 35 lipid–species variants (APOA5, ABCG5/8, BLK, LPL, FADS2, COL5A1, GALNT16, GLTPD2, MBOAT7 and SPTLC3) were associated with at least one of the CVD outcomes (FDR < 5%)

(Fig. 5; Supplementary Data 8). These included novel associations of variants at COL5A1 with cerebrovascular disease ($P = 4.6 \times 10^{-4}$), GALNT16 with angina ($P = 9.3 \times 10^{-4}$), MBOAT7 with venous thromboembolism ($P = 1.3 \times 10^{-3}$), GLTPD2 with atherosclerosis ($P = 5.3 \times 10^{-4}$) and SPTLC3 with intracerebral haemorrhage ($P = 1.0 \times 10^{-3}$) (Fig. 5). FADS1-2-3 is a well-known lipid modifying locus; however, like many other known lipid loci, its effects on CVD risk has been unclear. We found an association of FADS2 rs28456-G with peripheral artery disease ($P = 2.2 \times 10^{-4}$) and aterial embolism and thrombosis ($P = 2.5 \times 10^{-4}$). BLK (rs1478898-A) was also found to be associated with decreased risk of obesity (OR = 0.97, $P = 5.6 \times 10^{-8}$) and type 2 diabetes (OR = 0.96, $P = 4.5 \times 10^{-5}$).

Several studies have suggested a role for sphingolipids, including CERs and SMs, in the pathogenesis of CVDs. CER (d18:1/24:0) and CER (d18:1/24:1) have been reported to be associated with the increased risk of CVD events[9]. We found that the CER (d18:1/24:1) decreasing variant SPTLC3 rs364585-G was associated with decreased risk of intracerebral haemorrhage, while CER (d18:1/24:0) increasing variant ZNF385D rs13070110-C was nominally associated with increased risk of intracerebral haemorrhage. Furthermore, consistent with the observation

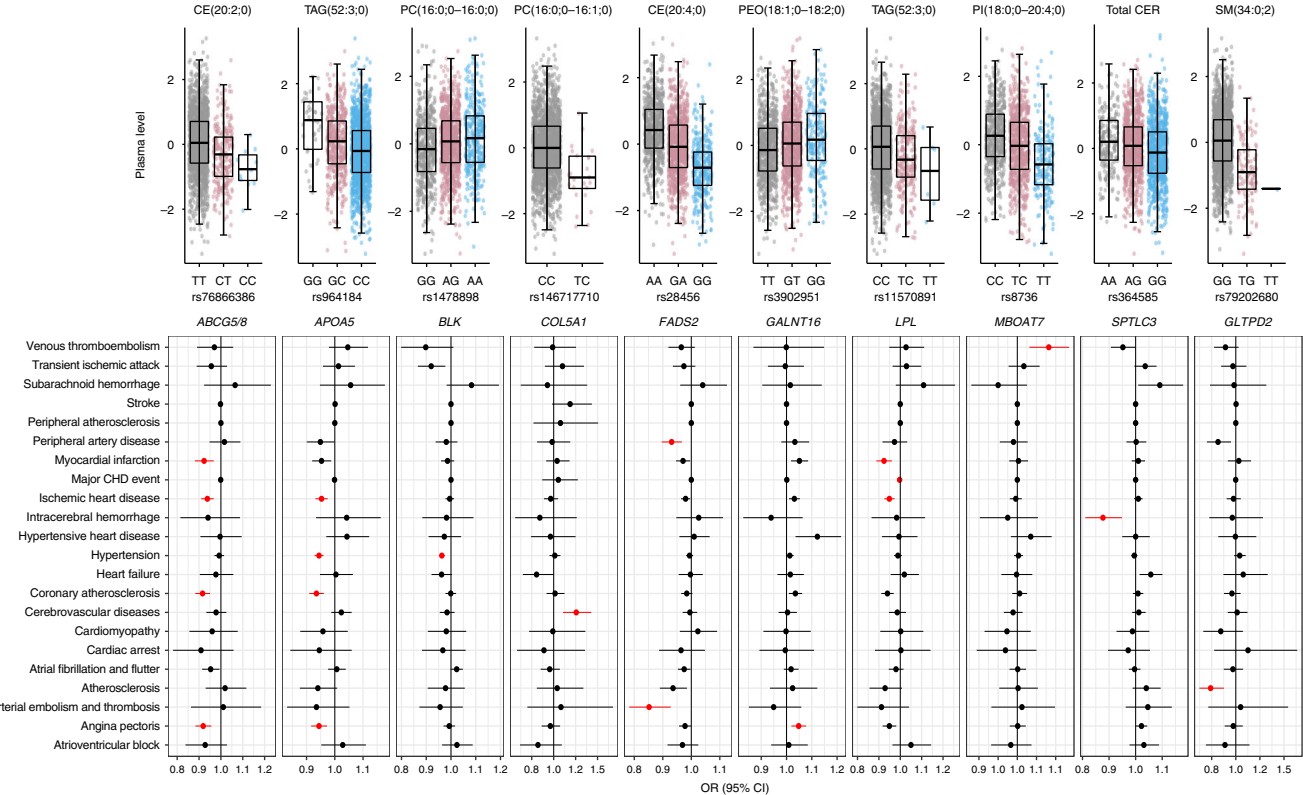

**Fig. 5** Relationship between lipid species-associated variants and risk of CVDs. The upper panel shows the association of the identified variants with the strongest associated lipid species. Boxplots show the interquartile range (IQR) defined by the bounds of the box with the median (middle line of the box) of plasma levels of the respective lipid species for each genotype of the variants; whiskers extend to the largest/smallest values no further than 1.5 times the IQR. The lower panel depicts the relationship between the identified variants with CVD phenotypes. The effect sizes (odds ratio) with 95% confidence interval are plotted with respect to the alternate alleles. The associations with CVD phenotypes highlighted in red colour are significant at FDR <0.05

that elevated plasma SMs levels are atherogenic[24], we identified association of *GLTPD2* rs79202680-T (associated with reduced levels of SMs) with reduced risk of atherosclerosis.

**Mechanistic insights into lipid variants**. Next, we determined if the detailed lipidomic profiles could provide new mechanistic insights into the role of known lipid variants in lipid biology. We present two examples of well-established lipid variants here. First is the fatty acid desaturase (*FADS*) gene cluster that has been consistently reported to be associated with omega-3 and omega-6 fatty acids levels with inverse effects on different PUFAs. Its mechanism, however, has not been fully deciphered. Here, we found that the *FADS2* rs28456-G was associated with increased levels of lipids with a C20:3 acyl chain and decreased levels of lipids with C20:4, C20:5 and C22:6 acyl chains (Supplementary Fig. 5). The rs28456-G is also an eQTL that increases *FADS2* expression while reduces the expression of *FADS1* [GTEx v7]. These data together explain the inverse relationship of *FADS2* variants with lipids containing different polyunsatureated fatty acids (PUFAs) (Fig. 6).

Another example is lipoprotein lipase (*LPL*). *LPL* codes for lipoprotein lipase that is the master lipolytic factor of TAGs in TAG-enriched chylomicrons and VLDL particles. We found that *LPL* rs11570891-T was associated with reduced levels of medium length TAGs (C50–C56), with strongest associations with TAG (52:3;0). This suggested that LPL enzyme might have different efficiency in hydrolysis of TAGs of different length. We explored this possibility by evaluating (1) the effect of *LPL* rs11570891-T on LPL enzymatic activity and (2) the relationship between LPL

activity and plasma levels of TAGs of different length, using post-heparin LPL measured in the EUFAM cohort. We found that *LPL* rs11570891-T (an eQTL increasing *LPL* expression) was associated with increased LPL activity, which in turn was associated with TAG species with stronger effect on medium length TAGs than other TAGs (Fig. 6). Consistent with a previous report by Rhee et al.[16], variant rs964184-C at *APOA5*, which codes for the activator that stimulates LPL-mediated lipolysis of TAG-rich lipoproteins and their remnants, also showed association with medium length TAGs (Fig. 6). These results provide first clues to the probable variable role of LPL and APOA5 in the hydrolysis of different TAG species.

Similarly, the association patterns of some of the newly mapped loci suggested their underlying functions. For example, *SYNGR1* rs186680008-C showed strongest associations with decreased levels of lipid species with C20:3 acyl chain from different lipid classes, including CEs, PCs and PCOs (Supplementary Fig. 5), suggesting its role in PUFA metabolism (Fig. 6). *PTPRN2* rs10281741-G and *MIR100HG* rs10790495-G showed associations with reduced levels of long polyunsaturated TAG species, suggesting their role in negative regulation of either elongation and desaturation of fatty acids or incorporation of long-chain unsaturated fatty acids during TAG biosynthesis.

**Lipidomics provide higher statistical power**. As intermediate phenotypes are known to provide more statistical power, we assessed whether the lipid species could help to detect genetic associations with greater power than traditional lipids using

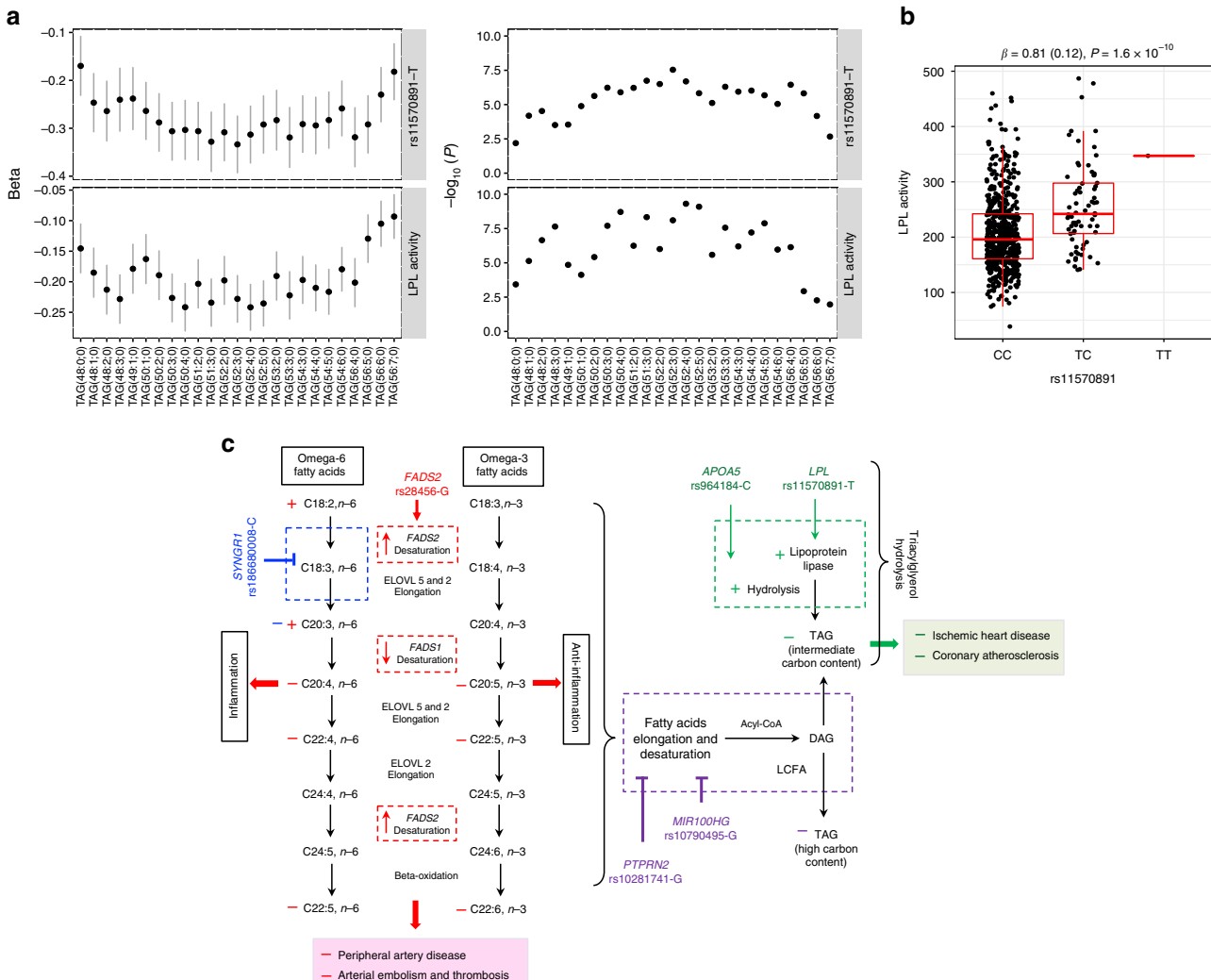

**Fig. 6** Patterns in associations and proposed mechanisms for the effect of identified variants on lipid metabolism and clinical outcomes. **a** Associations of *LPL* rs11570891-T and LPL activity with TAGs. Change (beta and standard errors) in plasma levels of TAGs per increase in standard deviation of LPL activity with their corresponding *P*-values, as calculated using linear regression model, are plotted in lower panel. The upper panel shows change (beta and standard errors) in plasma levels of TAGs per T allele with their corresponding *P*-values, as obtained from meta-analyses of genome-wide association analysis. **b** Association of *LPL* variant rs11570891 with LPL activity. The effect size (beta in standardised units and standard error in parenthesis) and *P*-value were calculated using linear mixed model. Boxplot depicts the interquartile range (IQR) defined by the bounds of the box, median (middle line) and whiskers extending to the largest/smallest values no further than 1.5 times the IQR. **c** Based on the patterns of the association of lipid species-associated loci with different lipid species, we propose that: (1) *LPL* rs11570891-T and *APOA5* rs964184-C might result in more efficient hydrolysis of medium length TAGs which might results in reduced CVD risk, (2) *FADS2* rs28456-G may have observed effect on PUFA metabolism through its inverse effect on *FADS2* and *FADS1* expressions, (3) *SYNGR1* rs18680008-C might have a role in the negative regulation of either desaturation of linoleic acid (C18:2,*n*-6) or elongation of gamma linoleic acid (C18:3,*n*-6). (4) *PTPRN2* rs10281741-G and *MIR100HG* rs10790495-G, which have very similar patterns of association with reduced level of long polyunsaturated TAGs, might have a role in negative regulation of either elongation and desaturation of fatty acids or incorporation of long chain unsaturated fatty acids in glycerol backbone during TAG biosynthesis. The positive (+) and negative (−) signs indicate increase or decrease, respectively, in level of lipid species or risk of disease as observed in our study, with different colours for different genetic variant

variants previously identified for traditional lipids (number of variants = 557; Supplementary Data 9). We found that molecular lipid species have much stronger associations than traditional lipids with the same sample size, except for well-known *APOE* and *CETP* (Fig. 7; Supplementary Data 10). The associations were several orders of magnitudes stronger for the variants in or near genes involved in lipid metabolism, such as *FADS1-2-3*, *LIPC*, *ABCG5/8*, *SGPP1* and *SPTLC3*. This shows that the lipidomics provides higher chances to identify lipid-modulating variants, particularly the ones with direct role in lipid metabolism, with much smaller sample size than traditional lipids.

## Discussion

We present findings from a large-scale study that integrate lipidome, genome and phenome revealing detailed description of genetic regulation of lipidome and its associations with CVD risk. In addition to enhancing the current understanding of genetic determinants of circulating lipids, our study highlights the potential of lipidomics in gene mapping for lipids and CVDs over traditional lipids. The study generates a publicly available knowledgebase of genetic associations of molecular lipid species and their relationships with thousands of clinical outcomes.

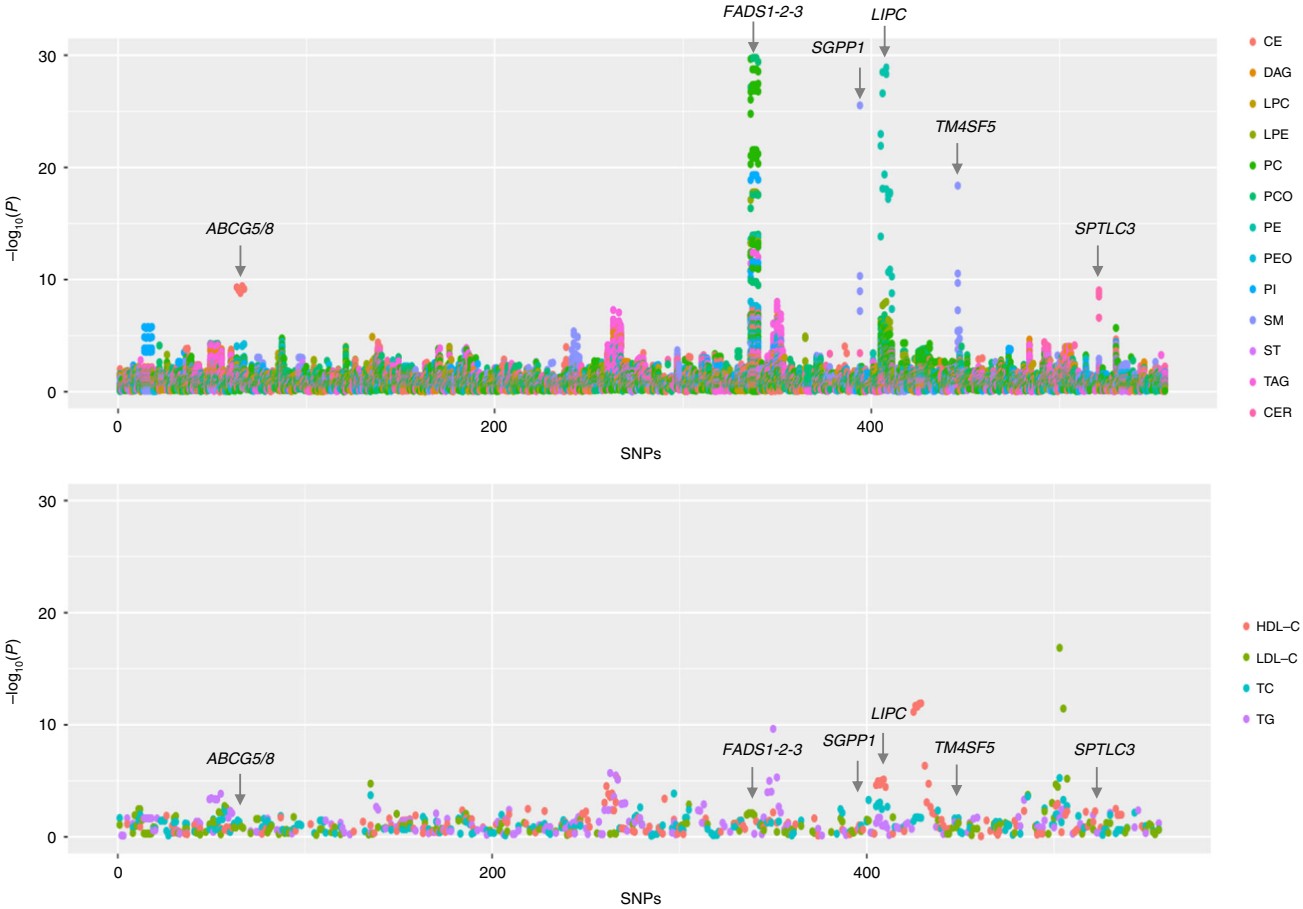

**Fig. 7** Association of known variants for traditional lipids with lipid species and traditional lipids. The *P*-values for the associations of the lead SNPs (557 SNPs available in our data set) identified through different genome-wide or exome-wide studies of traditional lipids (HDL-C, LDL-C, TG and TC) with lipid species (upper panel) and traditional lipids (lower panel) are plotted. The *y*-axis in the upper panel is capped at −log10 *P*-value = 30 for better representation of the data. The SNPs on the *x*-axis are serially arranged based on their chromosomal positions and as listed in the Supplementary Data 8. The points on the plots are colour coded by the lipid classes in the upper panel and traditional lipid in the lower panel. CER  ceramide, DAG  diacylglyceride, LPC  lysophosphatidylcholine, LPE  lysophosphatidylethanolamine, PC  phosphatidylcholine, PCO  phosphatidylcholine-ether, PE  phosphatidylethanolamine, PEO  phosphatidylethanolamine-ether, PI  phosphatidylinositol, CE  cholesteryl ester, SM  sphingomyelin, ST  sterol, TAG  triacyglycerol, TC total cholesterol, TG triglycerides

Despite the expected influence of dietary intake on the circulatory lipids, plasma levels of lipid species are found to be heritable, suggesting considerable role of endogenous regulation in lipid metabolism. Importantly, genetic mechanisms do not seem to regulate all lipid species in a lipid class in the same way, as also observed in recent mice lipidomics studies[25,26]. Longer and more unsaturated lipid species from different lipid classes clearly display stronger genetic correlations. These observations are consistent with a previous study based on family pedigrees[21]. Our finding is important in the light of the proposed role of lipids containing PUFAs in CVDs, diabetes and other disorders[27–29]. Identification of genetic factors regulating these particular lipids is important for understanding the subtleties of lipid metabolism and devising preventive strategies including dietary interventions. Our study provides multiple leads in this direction by identifying 11 genomic loci (*KLHL17*, *APOA5*, *CD33*, *SHTN1*, *FADS2*, *LIPC*, *MBOAT7*, *MIR100HG*, *PTPRN2*, *PDHA2* and *TMEM86B*) associated with long, polyunsaturated lipids at genome-wide significance. Of these, *FADS2*, *APOA5*, *LPL* and *MBOAT7* variants were also associated with risk of CVDs (Fig. 5).

Further, we mapped genetic variants for lipid species from several lipid classes, including CERs, CEs, TAGs, SMs and PCs, that are shown to predict CVD risk[4–9]. Our PheWAS analyses

also suggested relationship between many of the mapped genetic variants and CVD outcomes. This knowledge can directly fuel studies on CVD prediction or drug target discovery. For instance, CERs and CEs have also been reported to associate with increased risk of CVD events[5–9]. Our study revealed three loci associated with CEs, including *FADS2* and two novel loci-*ABCG5/8* and *SYNGR1*, and two loci for CERs (*SPTLC3* and *ZNF385D*). CER species, particularly CER (d18:1/24:0) and CER (d18:1/24:1) are recently reported to be associated with the increased risk of CVD[9]. We identified two variants near *SPTLC3* and *ZNF385D* that modulate the plasma levels of CER (d18:1/24:1) and CER (d18:1/24:0), respectively, and risk for intracerebral haemorrhage. This information could also guide future studies to establish the causal relationship between lipid species and CVD.

The detailed lipidomic profile also provided clues towards understanding the mechanisms of effects of well-established lipid loci like *FADS2* and *LPL* on lipid metabolism and CVD risks. We show how the inverse effects of *FADS2* rs28456-G on the expression of two desaturases (*FADS2* and *FADS1*) could explain its opposite effects on lipids with different PUFAs. The delta-6 desaturation by FADS2 generates gamma-linolenic acid and stearidonic acid that by elongation yield dihomo-gamma-linolenic acid and eicosatetraenoic acid (Fig. 6)[30]. Further,

delta-5 desaturation of dihomo-gamma-linolenic acid by *FADS1* generates arachidonic acid and eicosapentaenoic acid. Thus, as depicted in Fig. 6, the inverse effects of *FADS2* rs28456-G on *FADS2* and *FADS1* expressions explain its opposite effects on different PUFAs. The association of *FADS2* rs28456-G with the reduced levels of lipids containing arachidonic acid may also explain its association with reduced risk of atherosclerotic CVD outcomes—peripheral artery disease (PAD) and aterial embolism and thrombosis.

LPL and APOA5 are the key players in TAG hydrolysis. Our integrated approach suggested that their activity could be different for different TAG species with higher efficiency for medium length TAGs (C50–C56). We show that an *LPL* variant increases the LPL activity resulting in decreased levels of medium length TAGs. The association of the LPL variant with reduced susceptibility to CVD and type 2 diabetes could be mediated through the decrease in medium length TAGs (Fig. 5). This is consistent with a previous report that showed a similar pattern of association of levels of TAG species with type 2 diabetes[31].

Similarly, the patterns of assocations of newly mapped loci also suggested their involvement in the regulation of lipid metabolism. For example, rs10281741-G near *PTPRN2* and rs10790495-G near *MIR100HG* showed distinct association patterns with TAGs, with strongest association with long polyunsaturated TAGs. *PTPRN2* codes for protein tyrosine phosphatase receptor N2 with a possible role in pancreatic insulin secretion and development of diabetes mellitus[32], while *MIR100HG* rs10790495 is an eQTL for the heat-shock protein HSPA8 that has a role in cell proliferation[33]. However, it is not known if *PTPRN2* and *MIR100HG* or *HSPA8* have any role in lipid metabolism.

Finally, we show that lipidomic profiles capture information beyond traditional lipids and provide an opportunity to identify additional genetic variants influencing lipid metabolism and disease risk. Previously, Petersen et al. showed that lipoprotein subfractions correlate with traditional lipids and strengthen genetic associations at known lipid loci and that these loci explain more of the variance of lipoprotein subfractions than of serum lipids[34]. Similarly, our study demonstrates that molecular lipid species have stronger statistical power compared with traditional lipids at known lipid loci using the same sample size. However, in contrast to Petersen et al., we found that many of the lipid species, including LPCs and PCs that have previously been associated with incident coronary heart disease risk[4–6], have low phenotypic and genotypic correlations with traditional lipids. We also show that the known lipid variants for traditional lipids explain less of the variance of lipid species than traditional lipids. Altogether, as expected these results suggest that lipidomic profiles could provide novel information that could not be captured by traditional lipids and lipoprotein measurements.

Our study had some potential limitations. Though our study represents one of the largest genetic screen of lipidomic variation, larger cohorts are needed to achieve its full understanding. Blood samples for the EUFAM cohort were drawn after an overnight fast whereas the FINRISK cohort samples had varied fasting duration. This, however, does not seem to have substantial effect on the results and their interpretation as shown in Supplementary Data 11 and Supplementary Fig. 6. Moreover, a recent study by Rämö et al. also demonstrated similar lipidomic profiles for dyslipidemias from the EUFAM and FINRISK cohorts[35]. The UK Biobank cohort is reported to have a "healthy volunteer" effect[36], which may affect the PheWAS results, however, given the large sample size, this is unlikely to have a substantial effect on genetic association analyses. Furthermore, lipidomic profiles were measured in whole plasma, which does not provide information at the level of individual lipoprotein subclasses and limits our ability to gain detailed mechanistic insights. We also excluded poorly

detected lipid species to ensure high data quality that narrowed the spectrum of lipidomic profiles. Further advances in lipidomics platforms might help to capture more comprehensive and complete lipidomic profiles, including the position of fatty acyl chains in the glycerol backbone of TAGs and glycerophospholipids and detection of sphingosine-1-P species and several other species, that would allow to overcome these limitations.

In conclusion, our study demonstrates that lipidomics enables deeper insights into the genetic regulation of lipid metabolism than clinically used lipid measures, which in turn might help guide future biomarker and drug target discovery and disease prevention.

## Methods

**Subjects and clinical measurements**. The study included participants from the following cohorts: EUFAM, FINRISK, FinnGen and UK Biobank. The EUFAM (The European Multicenter Study on Familial Dyslipidemias in Patients with Premature Coronary Heart Disease) study cohort is comprised of the Finnish familial combined hyperlipidemia families[37]. The families in EUFAM study were identified via probands admitted to Finnish university hospitals with a diagnosis of premature coronary heart disease. The probands had premature coronary heart disease and high levels of the total cholesterol, triglycerides, or both (≥ 90th Finnish age-specific and sex-specific population percentile), or low HDL-C levels (≤ 10th percentile). Invitation was extended to all the family members and spouses of the probands if at least one first-degree relative of the proband had high levels of the total cholesterol, triglycerides, or both. Venous blood samples were obtained from all participants after overnight fasting. Triglycerides and total cholesterol were measured by enzymatic methods using an automated Cobas Mira analyser (Hoffman-La Roche, Basel, Switzerland)[37,38]. HDL-C was quantified by phosphotungstic acid/magnesium chloride precipitation procedures, and LDL-C was calculated using the Friedewald formula[39].

The Finnish National FINRISK study is a population-based survey conducted every 5 years since 1972, and thus far samples have been collected in 1992, 1997, 2002, 2007 and 2012[40]. Collections from the 1992, 1997, 2002, 2007 and 2012 surveys are stored in the National Institute for Health and Welfare /THL) Biobank. Lipidomic profiling was performed for 1142 participants that were randomly selected from the FINRISK 2012 survey (Supplementary Table 1). The participants were advised to fast for at least 4 h before the examination and to avoid heavy meals earlier during the day. Venous blood samples were obtained from all the participants and sera were separated. HDL-C, triglycerides and total cholesterol were measured with enzymatic methods (Abbott laboratories, Abbott Park, IL, USA) with Abbott Architect c8000 clinical chemistry analyser[40].

The FinnGen data release 2 is composed of 102,739 Finnish participants. The phenotypes were derived from ICD codes in Finnish national hospital registries and cause-of-death registry as a part of FinnGen project. The quality of the CVD diagnoses in these registers has been validated in previous studies[41–45]. The UK Biobank data is comprised of >500,000 participants based in UK and aged 40–69 years, annotated for over 2000 phenotypes[46]. The PheWAS analyses in this study included 408,961 samples from white British participants.

**Ethics statement**. The study was conducted in accordance with the principles of the Helsinki declaration. Written informed consent was obtained from all the study participants. The study protocols were approved by the ethics committees of the participating centres (The Hospital District of Helsinki and Uusimaa Coordinating Ethics committees, approval No. 184/13/03/00/12). For the Finnish Institute of Health and Welfare (THL) driven FinnGen preparatory project (here called FinnGen), all patients and control subjects had provided informed consent for biobank research, based on the Finnish Biobank Act. Alternatively, older cohorts were based on study specific consents and later transferred to the THL Biobank after approval by Valvira, the National Supervisory Authority for Welfare and Health. Recruitment protocols followed the biobank protocols approved by Valvira. The Ethical Review Board of the Hospital District of Helsinki and Uusimaa approved the FinnGen study protocol Nr HUS/990/2017. The FinnGen preparatory project is approved by THL, approval numbers THL/2031/6.02.00/2017, amendments THL/341/6.02.00/2018, THL/2222/6.02.00/2018 and THL/283/6.02.00/2019. All DNA samples and data in this study were pseudonymized.

**Lipidomic profiling**. Mass spectrometry-based lipid analysis of 2181 participants was performed in three batches-353 and 686 EUFAM participants in two batches and 1142 FINRISK participants in third batch at Lipotype GmbH (Dresden, Germany). Samples were analysed by direct infusion in a QExactive mass spectrometer (Thermo Scientific) equipped with a TriVersa NanoMate ion source (Advion Biosciences)[47]. The data were analysed using in-house developed lipid identification software based on LipidXplorer[48,49]. Post processing and normalisation of data were performed using an in-house developed data management system. Only lipids with signal-to-noise ratio >5 and amounts at least fivefold higher than in the corresponding blank samples were considered for further

analyses. Reproducibility of the assay was assessed by the inclusion of reference plasma samples (eight reference samples for EUFAM and three reference samples for FINRISK) per 96-well plate. Median coefficient of variation was <10% across all batches. The data were corrected for batch and drift effects. Lipid species detected in <80% of the samples in any of the batches and samples ($N = 64$) with low lipid contents were excluded. Among the lipid species which passed quality control, a total of 141 lipid species from 13 lipid classes (Supplementary Table 2) were detected consistently in all three batches and were included in all analysis. The total amounts of lipid classes were calculated by summing up the absolute concentrations of all lipid species belonging to each lipid class. The measured concentrations of the lipid species and calculated class total were transformed to normal distribution by rank-based inverse normal transformation.

It is to be noted that Lipotype platform used in the study detected many additional lipid species ($N = 83$) that were not captured previously by other platforms. The list of the lipid species detected by different platforms and overlaps across the platforms are provided in the Supplementary Data 12 and Supplementary Fig. 7.

**Genotyping and imputation**. Genotyping for both EUFAM and FINRISK cohorts was performed using the HumanCoreExome BeadChip (Illumina Inc., San Diego, CA, USA). The genotype calls were generated together with other available data sets using zCall at the Institute for Molecular Medicine Finland (FIMM). Genotype data underwent stringent quality control (QC) before imputation that included exclusion of samples with low call rate (<95%), sex discrepancies, excess heterozygosity and non-European ancestry. Variants with low call rate (<95%) and deviation from Hardy–Weinberg Equilibrium (HWE $P < 1 \times 10^{-6}$) were excluded. Imputation was performed using IMPUTE2[50], which used two population-specific reference panels of 2690 high-coverage whole-genome and 5093 high-coverage whole-exome sequence data. Variants with imputation info score <0.70 were filtered out. After QC on lipidomic profiles and imputed variants, all subsequent analyses included 2045 individuals and ~9.3 million variants with MAF >0.005 that were available in both cohorts.

FinnGen samples were genotyped with Illumina and Affymetrix arrays (Thermo Fisher Scientific, Santa Clara, CA, USA). Genotype calls were made with GenCall and zCall algorithms for Illumina and AxiomGT1 algorithm for Affymetrix chip genotyping data. Genotyping data produced with previous chip platforms were lifted over to build version 38 (GRCh38/hg38) following the protocol described here: dx.doi.org/10.17504/protocols.io.nqtddwn. Samples with sex discrepancies, high genotype missingness (> 5%), excess heterozygosity (+-4SD) and non-Finnish ancestry were removed. Variants with high missingness (> 2%), deviation from HWE ($P < 1e$-$6$) and low minor allele count (MAC < 3) were removed. Pre-phasing of genotyped data was performed with Eagle 2.3.5 (https://data.broadinstitute.org/alkesgroup/Eagle/) with the default parameters, except the number of conditioning haplotypes was set to 20,000. Imputation was carried out by using the population-specific SISu v3 imputation reference panel with Beagle 4.1 (version 08Jun17.d8b, https://faculty.washington.edu/browning/beagle/b4_1.html) as described in the following protocol: [dx.doi.org/10.17504/protocols.io.nmndc5e]. SISu v3 imputation reference panel was developed using the high-coverage (25–30x) whole-genome sequencing data generated at the Broad Institute of MIT and Harvard and at the McDonnell Genome Institute at Washington University; and jointly processed at the Broad Institute. Variant callset was produced with GATK HaplotypeCaller algorithm by following GATK best-practices for variant calling. Genotype-, sample- and variant-wise QC was applied in an iterative manner by using the Hail framework v0.1 [https://github.com/hail-is/hail]. The resulting high-quality WGS data for 3775 individuals were phased with Eagle 2.3.5 as described above. Post-imputation quality control involved excluding variants with INFO score < 0.7.

Genotyping for the majority of the UK Biobank participants was done using the Affymetrix UK Biobank Axiom Array, while a subset of participants was genotyped using the Affymetrix UK BiLEVE Axiom Array. Details about the quality control and imputation of UK Biobank cohort are described by Bycroft et al.[51].

**Heritability estimates and genetic correlations**. For heritability and genetic correlation estimation, rank-based inverse-transformed measures of lipid species, computed separately for the EUFAM and FINRISK cohorts, were combined to increase statistical power. The residuals of inverse-transformed measures after regressing for age, sex, first ten principal components (PCs) of genetic population structure, lipid medication, hormone replacement therapy, thyroid condition and type 2 diabetes were used as phenotypes. SNP-based heritability estimates were calculated using the variance component analysis using a genetic relationship matrix (GRM) as implemented in biMM[52]. Only the good quality variants with missingness <10% and MAF >0.005 were used to generate the GRM. The GRM was generated using GCTA by setting the off-diagonal elements that are <0.05 to 0 as proposed by Zaitlen et al.[53]. This allows to estimate SNP-based heritability in family data without removing closely related individuals. The heritability estimates of lipid species in different groups were compared using Wilcoxon rank-sum test.

The genetic correlation between each pair of lipid species and between each lipid species and traditional lipids was determined using the generated GRM with bivariate linear mixed model as implemented in biMM. The correlations based on the plasma levels (termed as phenotypic correlations) between all the pairs of the

lipid species and traditional lipids were calculated using Pearson's correlation coefficient. The heatmaps and hierarchical clustering based on genetic and phenotypic correlations were generated using heatmap.2 in R. As lipid-lowering medications could affect the plasma levels of lipid species, all analyses were adjusted for the usage of lipid-lowering medications, and separate analyses were also performed after excluding individuals using lipid-lowering medications ($N = 172$).

**Lipidomics GWAS**. We performed univariate association tests for 141 individual lipid species, 12 total lipid classes and 4 traditional lipid measures (HDL-C, LDL-C, total cholesterol and triglycerides), in all batches to control for possible batch effects and combined the summary statistics by meta-analysis. The association analyses for the EUFAM cohort were performed using linear mixed models, including the above-mentioned covariates as fixed effects and kinship matrix as random effect as implemented in MMM[54]. The kinship matrices for the GWAS analyses were computed separately for each chromosome to include the variants from the other chromosomes using directly genotyped variants with MAF >0.01 and missingness <2%. The FINRISK cohort was analysed with linear regression model adjusting for age, sex, first ten PCs, lipid medication and diabetes using SNPTEST v2.5[55]. Meta-analyses were performed using the inverse variance weighted method for fixed effects adjusted for genomic inflation factor in METAL[56]. In addition, analyses adjusting for the traditional lipids (in addition to above-mentioned covariates) were also performed for the identified variants to determine the independent effect on lipid species.

Test statistics were adjusted for $\lambda$ values if >1.0 before meta-analyses. Genomic inflation factor ($\lambda$) ranged from 0.98 to 1.19 across the batches whereas the final $\lambda$ values for meta-analysis ranged from 0.998 to 1.045 (Supplementary Data 13). The $P$-values obtained from the meta-analysis were considered to determine the SNP–lipid species associations. To account for multiple tests, the study-wide $P$-value threshold was set at <$1.5 \times 10^{-9}$ after correcting for 34 principal components (PCs) that explain over 90% of the variance in lipidomic profiles. Only the associations consistent in effect direction in all three batches were considered significant. Variants were designated as new if not located within 1 Mb of any previously reported variants for lipids (any of the traditional lipids and molecular lipid species); and as independent signal in known locus if located within 1 Mb but $r^2 < 0.20$ with the previous lead variants and confirmed by conditional analysis. Variants with the strongest association in the identified lipid species loci was identified as the lead variants, and were annotated to the nearest gene for the new loci.

**PheWAS**. We identified 25 CVD-related outcomes from the derived phenotypes in the FinnGen and UK Biobanks (Supplementary Table 3). Associations between the 35 lead variants from the identified loci and 25 selected CVD phenotypes in FinnGen cohort were obtained from the ongoing analyses as a part of the FinnGen project. The associations were tested using saddle point approximation method adjusting for age, sex and first 10 PCs as implemented in SPAtest R package[57]. Associations between selected binary phenotypes and 35 lead variants in UK Biobank were obtained from Zhou et al. that were tested using logistic mixed model in SAIGE with a saddle point approximation and adjusting for first four principal components, age and sex (https://www.leelabsg.org/resources)[58]. Data for four phenotypes were not available from Zhou et al. and hence were obtained from http://www.nealelab.is/uk-biobank/. Associations of quantitative traits were tested using linear regression models with the same covariates as mentioned above, both for Finnish and UK Biobank cohorts. Meta-analyses of both cohorts were performed using the inverse variance weighted method for fixed effects model in METAL. The $P$-values obtained from the meta-analyses of the two cohorts are reported for PheWAS associations. All the PheWAS associations with false discovery rate (FDR) <5% evaluated using the Benjamini–Hochberg method and consistent direction of effects were considered significant.

**Variance explained**. To determine the variance explained by the known loci for traditional lipids, we included all the lead variants with MAF >0.005 in 250 genomic loci that have previously been associated with one or more of the four traditional lipids. Of the 636 reported variants, 557 variants with MAF >0.005 (including six proxies) were available in our QC passed imputed genotype data (Supplementary Data 10). A genetic relationship matrix (GRM) based on these 557 variants was generated using GCTA that was used to determine the variance in plasma levels of all lipid species explained by the known variants using variance component analysis in biMM.

**LPL activity**. The post-heparin lipoprotein lipase (LPL) after 15 min of heparin load was measured for 630 individuals in the EUFAM cohort using the ELISA method developed by Antikainen et al.[59]. The measured values were transformed using rank-based inverse normal transformation. Associations between the LPL activity and plasma levels of TAGs were determined using linear regression model adjusted for age, sex, lipid medication, hormone replacement therapy, thyroid condition and type 2 diabetes. Association between the *LPL* variant rs11570891 and LPL activity was tested using linear mixed model adjusted for age, sex, first ten PCs of genetic population structure, lipid medication, hormone replacement therapy,

thyroid condition and type 2 diabetes as fixed effect and kinship matrix as random effect as implemented in MMM.

**Reporting summary**. Further information on research design is available in the Nature Research Reporting Summary linked to this article.

## Data availability

The full lipidomics GWAS summary level data are available on the web-based database [https://mqtl.fimm.fi]. Similarly, the PheWAS summary data can be obtained through [https://www.leelabsg.org/resources] and [http://www.nealelab.is/uk-biobank/]. The data presented in the figures and other summary level data are contained within the Supplementary Files and Supplementary Data. Other data are available through the Institute for Molecular Medicine Finland Data Access Committee on reasonable request after appropriate ethical approval.

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

## Acknowledgements

We would like to thank Sari Kivikko, Huei-Yi Shen and Ulla Tuomainen for management assistance. We thank all study participants of the study for their participation. The FINRISK and FinnGen data used for the research were obtained from THL Biobank. We thank the THL DNA laboratory for its skillful work to produce the DNA samples used in the genotyping work, which was used in this study. Part of the genotyping was performed by the Institute for Molecular Medicine Finland FIMM Technology Centre, University of Helsinki. This research has been conducted using the UK Biobank Resource with application number 22627. This work was supported by National Institutes of Health [grant HL113315 to S.R., M.R.T., N.F. and A.P.]; Finnish Foundation for Cardiovascular Research [to S.R., V.S., M.R.T., M.J. and A.P.]; Academy of Finland Center of Excellence in Complex Disease Genetics [grant 312062 to S.R.]; Academy of Finland [285380 to S.R., 288509 to M.P.]; Jane and Aatos Erkko Foundation [to M.J.]; Sigrid Jusélius Foundation [to S.R. and M.R.T.]; Horizon 2020 Research and Innovation Programme [grant 692145 to S.R.]; EU-project RESOLVE (EU 7th Framework Programme) [grant 305707 to M.R. T.]; HiLIFE Fellowship [to S.R.]; Helsinki University Central Hospital Research Funds [to M.R.T.]; Magnus Ehrnrooth Foundation [to M.J.]; Leducq Foundation [to M.R.T.]; Ida Montin Foundation [to P.R.]; MD-PhD Programme of the Faculty of Medicine, University of Helsinki [to J.T.R.]; Doctoral Programme in Population Health, University of Helsinki [to J.T.R. and P.R.]; Finnish Medical Foundation [to J.T.R.]; Emil Aaltonen Foundation [to J.T.R. and P.R.]; Biomedicum Helsinki Foundation [to J.T.R.]; Paulo Foundation [to J.T.R.]; Idman Foundation [to J.T.R.]; Veritas Foundation [to J.T.R.]; FIMM-EMBL PhD Fellowship grant [to S.H.]. The FinnGen project is funded by two grants from Business Finland (HUS 4685/31/2016 and UH 4386/31/2016) and nine industry partners (AbbVie, AstraZeneca, Biogen, Celgene, Genentech, GSK, Merck, Pfizer and Sanofi). The funders had no role in study design, data collection and analysis, decision to publish or preparation of the paper.

## Author contributions

R.T., S.R., M.R.T., K.S., A.P. and N.B.F. conceived and designed the study and wrote the paper; R.T. and J.T.R. performed the statistical analyses; J.T.K, M.K., J.K., T.T.J.K., A.S.H., H.L. and V.S. were involved in FinnGen phenotype definitions and PheWAS analyses; S.H., J.N. and P.P. performed the genotype imputation; M.J.G., C.K. and M.A.S. performed lipidomic profiling and processed the raw data; S.K.S. and M.P. provided critical inputs in statistical analyses; M.J. provided critical inputs in the interpretation of the data; P.R., S.S., N.M., N.O.S., M.J.D., V.S., N.B.F., A.P., M.R.T., K.S. and S.R. acquired the data. All authors read, revised and approved the paper.

## Additional information

**Competing interests:** V.S. has participated in a conference trip sponsored by Novo Nordisk and received an honorarium from the same source for participating in an advisory board meeting. M.J.G. is an employee of Lipotype GmbH, C.K. is a shareholder and employee of Lipotype GmbH, K.S. is a shareholder and CEO of Lipotype GmbH. M. A.S. is a shareholder of Lipotype GmbH and an employee of Łukasiewicz Research Network–PORT Polish Center for Technology Development. The remaining authors have no relevant competing interests.

Rubina Tabassum[1], Joel T. Rämö[1], Pietari Ripatti[1], Jukka T. Koskela[1], Mitja Kurki[1,2,3], Juha Karjalainen[1,4,5], Priit Palta[1,6], Shabbeer Hassan[1], Javier Nunez-Fontarnau[1], Tuomo T.J. Kiiskinen[1,7], Sanni Söderlund[8], Niina Matikainen[8,9], Mathias J. Gerl[10], Michal A. Surma[10,11], Christian Klose[10], Nathan O. Stitziel[12,13,14], Hannele Laivuori[1,15,16], Aki S. Havulinna[1,7], Susan K. Service[17], Veikko Salomaa[7], Matti Pirinen[1,18,19], FinnGen Project, Matti Jauhiainen[7,20], Mark J. Daly[1,4], Nelson B. Freimer[17], Aarno Palotie[1,4,21], Marja-Riitta Taskinen[8], Kai Simons[10,22] & Samuli Ripatti[1,4,23]

[1]Institute for Molecular Medicine Finland, HiLIFE, University of Helsinki, Helsinki, Finland. [2]Program in Medical and Population Genetics and Genetic Analysis Platform, Stanley Center for Psychiatric Research, Broad Institute of MIT and Harvard, Cambridge, MA, USA. [3]Psychiatric & Neurodevelopmental Genetics Unit, Massachusetts General Hospital, Boston, MA, USA. [4]Broad Institute of the Massachusetts Institute of Technology and Harvard University, Cambridge, MA, USA. [5]Analytic and Translational Genetics Unit, Massachusetts General Hospital and Harvard Medical School, Boston, MA, USA. [6]Estonian Genome Center, Institute of Genomics, University of Tartu, Tartu, Estonia. [7]National Institute for Health and Welfare, Helsinki, Finland. [8]Research Programs Unit, Diabetes & Obesity, University of Helsinki and Department of Internal Medicine, Helsinki University Hospital, Helsinki, Finland. [9]Endocrinology, Abdominal Center, Helsinki University Hospital, Helsinki, Finland. [10]Lipotype GmbH, Dresden, Germany. [11]Łukasiewicz Research Network-PORT Polish Center for Technology Development, Stablowicka 147 Str., 54-066 Wroclaw, Poland. [12]Cardiovascular Division, Department of Medicine, Washington University School of Medicine, Saint Louis, MO, USA. [13]Department of Genetics, Washington University School of Medicine, Saint Louis, MO, USA. [14]McDonnell Genome Institute, Washington University School of

Medicine, Saint Louis, MO, USA. [15]Department of Obstetrics and Gynecology, Tampere University Hospital and Tampere University, Faculty of Medicine and Health Technology, Tampere, Finland. [16]Medical and Clinical Genetics, University of Helsinki and Helsinki University Hospital, Helsinki, Finland. [17]Center for Neurobehavioral Genetics, Semel Institute for Neuroscience and Human Behavior, University of California, Los Angeles, CA, USA. [18]Department of Public Health, University of Helsinki, Helsinki, Finland. [19]Helsinki Institute for Information Technology HIIT and Department of Mathematics and Statistics, University of Helsinki, Helsinki, Finland. [20]Minerva Foundation Institute for Medical Research, Biomedicum, Helsinki, Finland. [21]Psychiatric & Neurodevelopmental Genetics Unit, Department of Psychiatry, Analytic and Translational Genetics Unit, Department of Medicine, and the Department of Neurology, Massachusetts General Hospital, Boston, MA, USA. [22]Max Planck Institute of Molecular Cell Biology and Genetics, Dresden, Germany. [23]Department of Public Health, Clinicum, Faculty of Medicine, University of Helsinki, Helsinki, Finland. A full list of consortium members appears at the end of the paper.

## FinnGen Project

Anu Jalanko[1], Jaakko Kaprio[1], Kati Donner[1], Mari Kaunisto[1], Nina Mars[1], Alexander Dada[1], Anastasia Shcherban[1], Andrea Ganna[1], Arto Lehisto[1], Elina Kilpeläinen[1], Georg Brein[1], Ghazal Awaisa[1], Jarmo Harju[1], Kalle Pärn[1], Pietro Della Briotta Parolo[1], Risto Kajanne[1], Susanna Lemmelä[1], Timo P. Sipilä[1], Tuomas Sipilä[1], Ulrike Lyhs[1], Vincent Llorens[1], Teemu Niiranen[7], Kati Kristiansson[7], Lotta Männikkö[7], Manuel González Jiménez[7], Markus Perola[7], Regis Wong[7], Terhi Kilpi[7], Tero Hiekkalinna[7], Elina Järvensivu[7], Essi Kaiharju[7], Hannele Mattsson[7], Markku Laukkanen[7], Päivi Laiho[7], Sini Lähteenmäki[7], Tuuli Sistonen[7], Sirpa Soini[7], Adam Ziemann[24], Anne Lehtonen[24], Apinya Lertratanakul[24], Bob Georgantas[24], Bridget Riley-Gillis[24], Danjuma Quarless[24], Fedik Rahimov[24], Graham Heap[24], Howard Jacob[24], Jeffrey Waring[24], Justin Wade Davis[24], Nizar Smaoui[24], Relja Popovic[24], Sahar Esmaeeli[24], Jeff Waring[24], Athena Matakidou[25], Ben Challis[25], David Close[25], Slavé Petrovski[25], Antti Karlsson[26], Johanna Schleutker[26], Kari Pulkki[26], Petri Virolainen[26], Lila Kallio[27], Arto Mannermaa[28], Sami Heikkinen[28], Veli-Matti Kosma[29], Chia-Yen Chen[30], Heiko Runz[30], Jimmy Liu[30], Paola Bronson[30], Sally John[30], Sanni Lahdenperä[30], Susan Eaton[30], Wei Zhou[4], Minna Hendolin[31], Outi Tuovila[31], Raimo Pakkanen[31], Joseph Maranville[32], Keith Usiskin[32], Marla Hochfeld[32], Robert Plenge[32], Robert Yang[32], Shameek Biswas[32], Steven Greenberg[32], Eija Laakkonen[33], Juha Kononen[33], Juha Paloneva[33], Urho Kujala[33], Teijo Kuopio[34], Jari Laukkanen[35], Eeva Kangasniemi[36], Kimmo Savinainen[36], Reijo Laaksonen[36], Mikko Arvas[37], Jarmo Ritari[37], Jukka Partanen[37], Kati Hyvärinen[37], Tiina Wahlfors[37], Andrew Peterson[38], Danny Oh[38], Diana Chang[38], Edmond Teng[38], Erich Strauss[38], Geoff Kerchner[38], Hao Chen[38], Hubert Chen[38], Jennifer Schutzman[38], John Michon[38], Julie Hunkapiller[38], Mark McCarthy[38], Natalie Bowers[38], Tim Lu[38], Tushar Bhangale[38], David Pulford[39], Dawn Waterworth[39], Diptee Kulkarni[39], Fanli Xu[39], Jo Betts[39], Jorge Esparza Gordillo[39], Joshua Hoffman[39], Kirsi Auro[39], Linda McCarthy[39], Soumitra Ghosh[39], Meg Ehm[39], Kimmo Pitkänen[40], Tomi Mäkelä[41], Anu Loukola[42], Heikki Joensuu[42], Juha Sinisalo[42], Kari Eklund[42], Lauri Aaltonen[42], Martti Färkkilä[42], Olli Carpen[42], Paula Kauppi[42], Pentti Tienari[42], Terhi Ollila[42], Tiinamaija Tuomi[42], Tuomo Meretoja[42], Anne Pitkäranta[42], Joni Turunen[42], Katariina Hannula-Jouppi[42], Sampsa Pikkarainen[42], Sanna Seitsonen[42], Miika Koskinen[42], Antti Palomäki[43], Juha Rinne[43], Kaj Metsärinne[43], Klaus Elenius[43], Laura Pirilä[43], Leena Koulu[43], Markku Voutilainen[43], Markus Juonala[43], Sirkku Peltonen[43], Vesa Aaltonen[43], Andrey Loboda[44], Anna Podgornaia[44], Aparna Chhibber[44], Audrey Chu[44], Caroline Fox[44], Dorothee Diogo[44], Emily Holzinger[44], John Eicher[44], Padhraig Gormley[44], Vinay Mehta[44], Xulong Wang[44], Johannes Kettunen[45], Katri Pylkäs[45], Marita Kalaoja[45], Minna Karjalainen[45], Reetta Hinttala[45], Riitta Kaarteenaho[45], Seppo Vainio[45], Tuomo Mantere[45], Seppo Vainio[46], Anne Remes[47], Johanna Huhtakangas[47], Juhani Junttila[47], Kaisa Tasanen[47], Laura Huilaja[47], Marja Luodonpää[47], Nina Hautala[47], Peeter Karihtala[47], Saila Kauppila[47], Terttu Harju[47], Timo Blomster[47], Hilkka Soininen[48], Ilkka Harvima[48], Jussi Pihlajamäki[48], Kai Kaarniranta[48], Margit Pelkonen[48], Markku Laakso[48], Mikko Hiltunen[48], Mikko Kiviniemi[48], Oili Kaipiainen-Seppänen[48], Päivi Auvinen[48], Reetta Kälviäinen[48], Valtteri Julkunen[48], Anders Malarstig[49], Åsa Hedman[49], Catherine Marshall[49], Christopher Whelan[49], Heli Lehtonen[49], Jaakko Parkkinen[49], Kari Linden[49], Kirsi Kalpala[49], Melissa Miller[49], Nan Bing[49],

Stefan McDonough[49], Xing Chen[49], Xinli Hu[49], Ying Wu[49], Annika Auranen[50], Airi Jussila[50], Hannele Uusitalo-Järvinen[50], Hannu Kankaanranta[50], Hannu Uusitalo[50], Jukka Peltola[50], Mika Kähönen[50], Pia Isomäki[50], Tarja Laitinen[50], Teea Salmi[50], Anthony Muslin[51], Clarence Wang[51], Clement Chatelain[51], Ethan Xu[51], Franck Auge[51], Kathy Call[51], Kathy Klinger[51], Marika Crohns[51], Matthias Gossel[51], Kimmo Palin[52], Manuel Rivas[53], Harri Siirtola[54] & Javier Gracia Tabuenca[54]

[24]Abbvie, Chicago, IL, USA. [25]Astra Zeneca, Cambridge, UK. [26]Auria Biobank, University of Turku and Hospital District of Southwest Finland, Turku, Finland. [27]Auria Biobank, Turku, Finland. [28]Biobank of Eastern Finland, University of Eastern Finland and Northern Savo Hospital District, Kuopio, Finland. [29]Biobank of Eastern Finland, Kuopio, Finland. [30]Biogen, Cambridge, MA, USA. [31]Business Finland, Helsinki, Finland. [32]Celgene, Summit, NJ, USA. [33]Central Finland Biobank, University of Jyväskylä and Central Finland Health Care District, Jyväskylä, Finland. [34]Central Finland Biobank, Jyväskylä, Finland. [35]Central Finland Health Care District, Jyväskylä, Finland. [36]Finnish Clinical Biobank Tampere, University of Tampere and Pirkanmaa Hospital District, Tampere, Finland. [37]Finnish Red Cross Blood Service and Finnish Hematology Registry and Clinical Biobank, Helsinki, Finland. [38]Genentech, San Francisco, CA, USA. [39]GlaxoSmithKline, Brentford, UK. [40]Helsinki Biobank, Helsinki, Finland. [41]HiLIFE, University of Helsinki, Helsinki, Finland. [42]Hospital District of Helsinki and Uusimaa, Helsinki, Finland. [43]Hospital District of Southwest Finland, Turku, Finland. [44]Merck, Kenilworth, NJ, USA. [45]Northern Finland Biobank Borealis, University of Oulu and Northern Ostrobothnia Hospital District, Oulu, Finland. [46]Northern Finland Biobank Borealis, Oulu, Finland. [47]Northern Ostrobothnia Hospital District, Oulu, Finland. [48]Northern Savo Hospital District, Kuopio, Finland. [49]Pfizer, New York, NY, USA. [50]Pirkanmaa Hospital District, Tampere, Finland. [51]Sanofi, Paris, France. [52]University of Helsinki, Helsinki, Finland. [53]University of Stanford, Stanford, CA, USA. [54]University of Tampere, Tampere, Finland

