## [Peer Review File · Nature Communications]

Reviewers' comments:

Reviewer #1 (Remarks to the Author):

This paper describes a GWAS with a new commercial lipidomics platform (Lipotype GmbH, Dresden, Germany). The platform covers 141 lipid species from 13 lipid classes. The sample size is substantial for a lipidomics GWAS (N=2,181). Variants were declared significant if the association with a lipid reached genome-wide significance ($p < 5E-8$) and if it showed consistent directionality in the three analyzed batches. At this significance level 35 lipid-species associations were detected. The authors discuss overlap with CVD risk. Overall this is an interesting study that provides new insights into the genetic architecture of lipid metabolism that merits publication.

I have a few specific remarks that may help improve the manuscript (some are personal taste): The authors state that this is the first large-scale GWAS of the lipidome. This is not entirely correct. The study of Illig et al. (Nat. Gen. 2010) with the Biocrates platform covered many lipid species in a targeted assay, followed by other studies using the same or an extended version of the platform. Furthermore, Shin et al. (Nat. Gen. 2014, see <http://gwas.eu>) and Long et al. (Nat. Gen. 2017, see Suppl Tab 1) used the Metabolon platform, which also covers a large set of lipid species. For instance, the claim that this study identified new locus-lipid species associations at previously reported lipid loci is not always correct. For instance, rs261290 near LIPC has been reported in association with 1-stearoyl-2-arachidonoyl-GPE (18:0/20:4) at a p-value of $1.68E-23$ (see for instance <http://snipa.org> using the block annotation or <http://www.phenoscanter.medschl.cam.ac.uk> with rs261290). A more stringent comparison with previous mGWAS should be provided (the phenoscanner API could be useful here).

Also, a more direct comparison of the platform content with previous studies (Metabolon, Biocrates) would be of great interest (maybe a Venn diagram, metabolites from Long et al. are in Supp Tab 1). It would also be important to know how the Lipotype platform fares on identical lipids and identical SNPs discovered by the Biocrates and Metabolon GWAS.

For the novel loci, regional association plots should be provided

The term "genetic sharing" is not well defined – do you mean overlapping associations on a same SNP with several phenotypes? A more stringent definition of genetic signal colocalization should be used.

The term "traditional lipids" should be introduced as referring to HDL, LDL, TC, and bulk TGs.

The expectation that lipidomics loci should overlap with "traditional lipid loci" should be revisited. This may be correct for triglycerides, but HDL and LDL particles are not lipids, only lipid containing particles. The authors write: "This implies that genetic variants associated with traditional lipids fail to capture the variations in plasma levels of many lipid subspecies" (page 6) ... why should these variants capture variations in plasma levels of many lipid subspecies? In this context the author may consider comparing their results to Petersen et al. [PMID: 22156577] (a suggestion, not a requirement).

I am personally not a big fan of heritability calculations (but others are, so I am not asking to remove them) – especially in the case of the FADS-associated lipids, the signal may come from a signal variant. It would be interesting to see how the heritability changes when the FADS-signal is regressed out (but this is just my curiosity, not a request I expect the authors to follow).

Page 6: I suggest deleting "Moreover, lipid species with genome-wide significant association had higher heritability estimates compared to the lipid species with no significant association", as this is evident, and at this point genome-wide significance has not been formally introduced.

The PheWeb database is a great addition to the paper – the link should be replaced by a domain name to be persistent after publication (not an IP address, which may change). Some of the links to the Oxford Big server do not work – the alleles are inverted.

The association of the ROCK1 variant with CVD appears stretched if there is no genetic signal in large CVD GWAS. The association of BLK with diabetes also appears quite stretched ($P=4.5E-5$) and should be substantiated ... the same holds for GALNT16 and LPL ... in this context would it be interesting to be more formal about how many tests with disease have been made.

Figure 2D: Delete Y-axis legend "Height"
Figure 4B: X-axis legend is too small

Reviewer #2 (Remarks to the Author):

The authors report results of the hitherto largest genome-wide study of lipidomic profiles. More specifically, they studied 141 lipid species, followed by a phenome-wide scan with 26 CVD related phenotypes. They identified 35 loci, 15 of them not reported with lipid-related phenotypes, yet. 10 of the loci were also related with CVD endpoints. Furthermore, they also showed highly heterogeneous estimates of heritability and variance explained.

Altogether, this is a very interesting, timely manuscript, which has been very thoroughly performed with state-of the art methods.

I only have one major point:

With individual GWAS on 141 lipid species and 4 traditional lipid measures, defining a p-value of smaller than 5×10^{-5} as genome-wide significant is not really appropriate. Of course, the correlation between the traits has to be taken into account, e.g. by correcting for principal components explaining the majority of the correlation between traits, as has been done in recently published metabolomics GWAS.

Some additional minor points:

Please check the numbering of the Supplementary Tables in the text, e.g. on page 20, it should probably be Suppl. Table 6 instead of 4, page 5: Suppl. Table 2 instead of 1.

Figure 3: what is the rationale of ordering?

All Figures and supplementary figures: Make sure they are readable in the final submitted version (maybe enlarge axis labels, if possible), e.g. not readable at the moment: y-axis in Figure 4 B, figure 6 A and B, suppl. Figure 5 and 6

Response to Referees

Reviewer #1 comments:

This paper describes a GWAS with a new commercial lipidomics platform (Lipotype GmbH, Dresden, Germany). The platform covers 141 lipid species from 13 lipid classes. The sample size is substantial for a lipidomics GWAS (N=2,181). Variants were declared significant if the association with a lipid reached genome-wide significance ($p < 5E-8$) and if it showed consistent directionality in the three analyzed batches. At this significance level 35 lipid-species associations were detected. The authors discuss overlap with CVD risk. Overall this is an interesting study that provides new insights into the genetic architecture of lipid metabolism that merits publication. I have a few specific remarks that may help improve the manuscript (some are personal taste):

We thank the reviewer for the kind words and valuable suggestions that have helped to greatly improve the manuscript.

1. The authors state that this is the first large-scale GWAS of the lipidome. This is not entirely correct. The study of Illig et al. (Nat. Gen. 2010) with the Biocrates platform covered many lipid species in a targeted assay, followed by other studies using the same or an extended version of the platform. Furthermore, Shin et al. (Nat. Gen. 2014, see <http://gwas.eu>) and Long et al. (Nat. Gen. 2017, see Suppl Tab 1) used the Metabolon platform, which also covers a large set of lipid species.

We regret the confusing statement. We acknowledge the fact that there have been efforts to identify genetic variants for lipid species previously, and we have mentioned and cited in the introduction section of the manuscript. We meant to imply that this is the first study integrating lipidome, genome and phenome at this scale. We have rephrased the statement to avoid any confusion in the revised manuscript as:

Page 10, Lines 24-25: *“We present findings from a large-scale study that integrate lipidome, genome and phenome revealing detailed description of genetic regulation and genetic architecture of the lipidome, and their associations with CVD risk.”*

2. For instance, the claim that this study identified new locus-lipid species associations at previously reported lipid loci is not always correct. For instance, rs261290 near LIPC has been reported in association with 1-stearoyl-2-arachidonoyl-GPE (18:0/20:4) at a p-value of $1.68E-23$ (see for instance <http://snipa.org> using the block annotation or <http://www.phenoscanter.medschl.cam.ac.uk> with rs261290). A more stringent comparison with previous mGWAS should be provided (the phenoscanter API could be useful here).

We thank the reviewer for pointing this out. We have now formally compared our results with the previous metabolomics GWAS results. As suggested, we checked for the previous

associations with lipid related metabolites at 35 lipid species associated loci identified in our study using block annotation in the SNI PA (<http://snipa.org>) and the PhenoScanner v2 (<http://www.phenoscanter.medschl.cam.ac.uk/>) and manually curated to include associations from literature search.

Out of the 35 identified lipid species associated loci, 11 loci were reported to associate with at least one of the lipid related metabolites analyzed in any previous mGWAS at $P < 5.0 \times 10^{-8}$. The Supplementary Table 5 has now been updated based on this information and the detailed list of all previous associations is also provided in a new supplementary table (Supplementary Table 6).

The text in the manuscript has been modified accordingly as:

Page 7, Lines 3-11: *“We also replicated the previous associations of FADS2, SYNE2, LIPC, LASS4 and MBOAT7 with the same lipid species¹³⁻²⁰. The previously reported associations at the known loci identified in previous metabolomics GWASs are provided in Supplementary Table 6. This information was obtained from the databases- SNI PA (<http://snipa.org>) using block annotation and PhenoScanner v2 (<http://www.phenoscanter.medschl.cam.ac.uk/>) and were manually curated to include associations from literature search. In addition, we also identified new locus-lipid species associations at previously reported lipid loci including new associations of variants at ABCG5/8 with CE(20:2;0) ($P=3.9 \times 10^{-10}$), MBOAT7 with PI(18:0;0-20:3;0) ($P=3.0 \times 10^{-12}$) and GLTPD2 with SM(34:0;) ($P=3.4 \times 10^{-22}$) (Supplementary Tables 4 and 5).”*

3. Also, a more direct comparison of the platform content with previous studies (Metabolon, Biocrates) would be of great interest (maybe a Venn diagram, metabolites from Long et al. are in Supp Tab 1).

We agree with the reviewer and believe that this would be an important information that would allow comparison of the results across the studies. We have now provided the list of the lipid species detected by different platforms in the Supplementary Table 14 and their overlaps in Supplementary Figure 8. The text in the revised manuscript reads as:

Page 16, Lines 20-23: *“It is to be noted that Lipotype platform used in the study detected many additional lipid species (N=83) that were not captured previously by other platforms. The list of the lipid species detected by different platforms and overlaps across the platforms are provided in the Supplementary Table 14 and Supplementary Figure 8.”*

To compare different platform content, we obtained lipid species detected by Metabolon (139 as reported by Long et al. 2017), Biocrates (107 as reported by Illig et al. 2013) and Lipotype (141 in this study). The direct comparison of the lipid species captured by different platforms is not straightforward as different platforms detect lipid species at different molecular levels i.e. at species [eg. PC(36:3;0)] or subspecies [eg. PC(18:1;0-18:2;0)] levels. Nevertheless, we identified the total lipid moieties captured by these three platforms by their species or

subspecies (whichever available) and looked for the identical match in the three platforms. A total of 296 lipid metabolites were found, of which Metabolon captured 121, Lipotype captured 141 and Biocrates captured 107 lipid moieties.

The overlaps among the platform at different levels are shown in the venn diagrams below (included as Supplementary Figure 8 in the revised manuscript). As shown in panel A, over 40% of the lipid species in Metabolon and Lipotype overlap with each other. There is little overlap with Biocrates as it does not inform about the acyl chain composition for most of the lipids. Panel b shows that overlap at species level i.e. when for instance PC(34:0;0) and PC(16:0;0-18:0;0) were considered as match at species level. If we only consider the lipid species with acyl chain composition information in all three platforms (panel c), almost 50% of Biocrates lipid species could be detected by both Metabolon and Lipotype.

Figure: Comparison of lipid species detected by different platforms. a) Comparison of all the lipid moieties identified; b) Comparison at species level; c) Comparison at subspecies level.

4. It would also be important to know how the Lipotype platform fares on identical lipids and identical SNPs discovered by the Biocrates and Metabolon GWAS.

We thank the reviewer for this valuable suggestion. We have now summarized the results for all the variants previously identified in mGWAS and also the associations between the identical lipids and identical SNPs in the revised manuscript. The text in the manuscript reads as:

Page 7, Lines 12-18: *“Further, we systematically evaluated the associations of variants previously identified in metabolomics GWAS (126 variants from 46 loci available in our dataset out of 132 reported) with 141 lipid species. Of these known variants, 76 variants from 12 loci showed association with 98 different lipid species with $P < 3.2 \times 10^{-5}$ (correcting for 46 loci and 34 PCs for lipid species) (Supplementary Table 7). Of the 134 previously reported variant-lipid species pair associations that could be examined in our dataset, 94 of such associations were replicated with the same direction of effect with $P < 3.7 \times 10^{-4}$ (accounting for 134 comparisons) in our study (Supplementary Table 8).”*

For this comparison, we obtained the list of all the published GWAS for metabolomics from <http://www.metabolomix.com/list-of-all-published-gwas-with-metabolomics/> and the associations with lipid related metabolites were extracted from the respective manuscripts. A total of 132 SNPs were found to be associated with at least one of the lipid related metabolites in these studies. Of these 132 SNPs, 126 were available in our dataset and a total of 134 identical SNP-lipid species pairs associations were available in our dataset. Kindly note that the effect sizes are not comparable across the studies as they differ in their units, however irrespective of that, we observed strong correlation in the effect sizes between previous studies and our study.

5. For the novel loci, regional association plots should be provided.

We have now included regional association plots for all 35 loci as Supplementary Figure 4.

6. The term “genetic sharing” is not well defined – do you mean overlapping associations on a same SNP with several phenotypes? A more stringent definition of genetic signal colocalization should be used.

We determined the pairwise genetic correlation for all the lipid species. Genetic sharing here implies that the two lipid species with high genetic correlation would likely have overlapping genetic variants associated with them. In the present study, we wanted to determine the extent of genetic sharing between the lipid species. We did not intend to formally determine the genetic variants that colocalize or overlap between the lipid species as this would have been very intensive and out of the scope and focus of the study.

To avoid confusion, we have replaced the term “genetic sharing” by “genetic correlation” at appropriate places.

7. The term “traditional lipids” should be introduced as referring to HDL, LDL, TC, and bulk TGs.

The text has been modified as:

Page 4, Lines 6-7: “*Standard lipid profiling measures traditional lipids (referred to LDL-C, HDL-C, total triglycerides, and total cholesterol) but...*”

8. The expectation that lipidomics loci should overlap with “traditional lipid loci” should be revisited. This may be correct for triglycerides, but HDL and LDL particles are not lipids, only lipid containing particles. The authors write: “This implies that genetic variants associated with traditional lipids fail to capture the variations in plasma levels of many lipid subspecies” (page 6) ... why should these variants capture variations in plasma levels of many lipid subspecies? In this context the author may consider comparing their results to Petersen et al. [PMID: 22156577] (a suggestion, not a requirement).

We agree with the reviewer and that is exactly what we want to formally demonstrate and emphasize with the data at hands. We intend to show that lipidomic profiles capture information beyond traditional lipids and provide an opportunity to identify additional genetic variants influencing lipid metabolism and disease risk. We have modified the section dealing with this in the revised manuscript. Moreover, as suggested we have also discussed our results in the light of previous observations by Petersen et al.

The modified text in the manuscript now read as:

Page 13, Lines 7-19: *“Finally, we show that lipidomic profiles capture information beyond traditional lipids and provide an opportunity to identify additional genetic variants influencing lipid metabolism and disease risk. Previously, Petersen et al. showed that lipoprotein subfractions correlate with traditional lipids and strengthen genetic associations at known lipid loci and that these loci explain more of the variance of lipoprotein subfractions than of serum lipids³⁴. Similarly, our study demonstrate that molecular lipid species have stronger statistical power compared to traditional lipids at known lipid loci using the same sample size. However, in contrast to Petersen et al., we found that many of the lipid species, including LPCs and PCs that have previously been associated with incident coronary heart disease risk⁴⁻⁶, have low phenotypic and genotypic correlations with traditional lipids. We also show that the known lipid variants for traditional lipid measures explain less of the variance of lipid species than traditional lipid measures. Altogether, as expected these results suggest that lipidomic profiles could provide novel information that could not be captured by traditional lipids and lipoprotein measurements.”*

9. I am personally not a big fan of heritability calculations (but others are, so I am not asking to remove them) – especially in the case of the FADS-associated lipids, the signal may come from a signal variant. It would be interesting to see how the heritability changes when the FADS-signal is regressed out (but this is just my curiosity, not a request I expect the authors to follow).

As suggested, we calculated the heritability estimates after excluding +/- 2.5Mb region flanking *FADS2* variant. We observed only a slight decrease in the estimates for most of the lipid species as shown in table below.

Though the FADS region variants are consistently shown to be associated with lipids, these variants are quite common in the population (~ 40%) and hence are unlikely to have large effect on the lipid levels. Moreover, as we mentioned in the manuscript, we identified 11 loci that were associated with polyunsaturated lipids. Hence, as expected, the genetic regulation of these lipid species seems to be quite polygenic in nature and a single locus does not contribute much to the total variance.

Table 1: Heritability estimates after excluding the FADS region

Species	Heritability estimates	SE	Heritability estimates excluding FADS2 region	SE
CE(14:0:0)	0.253	0.061	0.235	0.059
CE(15:0:0)	0.349	0.058	0.349	0.058
CE(16:0:0)	0.245	0.061	0.229	0.060
CE(16:1:0)	0.212	0.061	0.198	0.060
CE(17:0:0)	0.327	0.055	0.327	0.055
CE(17:1:0)	0.344	0.058	0.346	0.057
CE(18:0:0)	0.381	0.060	0.385	0.059
CE(18:1:0)	0.241	0.059	0.229	0.057
CE(18:2:0)	0.264	0.060	0.257	0.058
CE(18:3:0)	0.294	0.062	0.293	0.062
CE(20:2:0)	0.291	0.066	0.290	0.066
CE(20:3:0)	0.265	0.067	0.255	0.066
CE(20:4:0)	0.461	0.060	0.424	0.060
CE(20:5:0)	0.364	0.062	0.345	0.062
CE(22:6:0)	0.412	0.064	0.420	0.065
DAG(16:0:0-18:1:0)	0.308	0.070	0.309	0.070
DAG(18:1:0-18:1:0)	0.245	0.071	0.290	0.072
DAG(18:1:0-18:2:0)	0.252	0.072	0.253	0.072
TAG(48:0:0)	0.299	0.066	0.299	0.066
TAG(48:1:0)	0.305	0.066	0.306	0.066
TAG(48:2:0)	0.293	0.067	0.295	0.067
TAG(48:3:0)	0.237	0.068	0.238	0.068
TAG(49:1:0)	0.310	0.066	0.311	0.066
TAG(50:1:0)	0.294	0.069	0.295	0.069
TAG(50:2:0)	0.253	0.068	0.254	0.068
TAG(50:3:0)	0.232	0.068	0.233	0.068
TAG(50:4:0)	0.219	0.069	0.220	0.069
TAG(51:2:0)	0.318	0.067	0.319	0.067
TAG(51:3:0)	0.252	0.066	0.253	0.066
TAG(52:2:0)	0.288	0.070	0.290	0.070
TAG(52:3:0)	0.247	0.070	0.248	0.070
TAG(52:4:0)	0.228	0.069	0.228	0.069
TAG(52:5:0)	0.238	0.071	0.239	0.071
TAG(53:2:0)	0.366	0.067	0.367	0.067
TAG(53:3:0)	0.262	0.067	0.263	0.067
TAG(54:3:0)	0.240	0.069	0.241	0.069
TAG(54:4:0)	0.203	0.067	0.204	0.067
TAG(54:5:0)	0.229	0.069	0.230	0.069
TAG(54:6:0)	0.291	0.074	0.291	0.074
TAG(56:4:0)	0.308	0.076	0.309	0.076
TAG(56:5:0)	0.322	0.074	0.322	0.074
TAG(56:6:0)	0.263	0.069	0.263	0.069
TAG(56:7:0)	0.337	0.070	0.336	0.070
PC(14:0:0-16:0:0)	0.227	0.058	0.227	0.058
PC(14:0:0-18:1:0)	0.328	0.060	0.328	0.060
PC(16:0:0-16:0:0)	0.132	0.057	0.123	0.056
PC(16:0:0-16:1:0)	0.116	0.057	0.122	0.058
PC(16:0:0-17:1:0)	0.284	0.062	0.284	0.062
PC(16:0:0-18:0:0)	0.193	0.060	0.190	0.058
PC(16:0:0-18:1:0)	0.160	0.058	0.153	0.057
PC(16:0:0-18:2:0)	0.221	0.063	0.216	0.062
PC(16:0:0-18:3:0)	0.204	0.064	0.204	0.064
PC(16:0:0-20:1:0)	0.166	0.056	0.166	0.056
PC(16:0:0-20:2:0)	0.163	0.064	0.162	0.064
PC(16:0:0-20:3:0)	0.142	0.065	0.136	0.065
PC(16:0:0-20:4:0)	0.352	0.063	0.319	0.061
PC(16:0:0-20:5:0)	0.321	0.063	0.297	0.063
PC(16:0:0-22:4:0)	0.184	0.063	0.182	0.063
PC(16:0:0-22:5:0)	0.351	0.065	0.337	0.064
PC(16:0:0-22:6:0)	0.323	0.066	0.328	0.066
PC(16:1:0-18:1:0)	0.172	0.057	0.178	0.057
PC(16:1:0-18:2:0)	0.181	0.056	0.179	0.056
PC(17:0:0-18:2:0)	0.334	0.061	0.333	0.061
PC(17:0:0-20:3:0)	0.419	0.071	0.420	0.071
PC(17:0:0-20:4:0)	0.539	0.062	0.536	0.062
PC(18:0:0-18:1:0)	0.232	0.057	0.239	0.057
PC(18:0:0-18:2:0)	0.334	0.063	0.333	0.062
PC(18:0:0-18:3:0)	0.244	0.064	0.244	0.064
PC(18:0:0-20:2:0)	0.348	0.064	0.346	0.064
PC(18:0:0-20:3:0)	0.261	0.067	0.273	0.068
PC(18:0:0-20:4:0)	0.471	0.061	0.452	0.062
PC(18:0:0-20:5:0)	0.297	0.064	0.293	0.064
PC(18:0:0-22:5:0)	0.299	0.066	0.297	0.066

PC(18:1;0-18:1;0)	0.272	0.058	0.268	0.058
PC(18:1;0-18:2;0)	0.312	0.061	0.300	0.061
PC(18:1;0-20:3;0)	0.175	0.059	0.185	0.060
PC(18:1;0-20:4;0)	0.367	0.064	0.331	0.063
PC(18:2;0-18:2;0)	0.257	0.061	0.254	0.061
PCO(16:0;0-16:0;0)	0.281	0.063	0.278	0.062
PCO(16:0;0-16:1;0)	0.205	0.067	0.206	0.067
PCO(16:0;0-18:1;0)	0.254	0.063	0.264	0.063
PCO(16:0;0-18:2;0)	0.287	0.061	0.300	0.061
PCO(16:0;0-20:3;0)	0.145	0.057	0.151	0.058
PCO(16:0;0-20:4;0)	0.309	0.062	0.313	0.062
PCO(16:1;0-16:0;0)	0.319	0.062	0.319	0.062
PCO(16:1;0-18:1;0)	0.361	0.062	0.363	0.061
PCO(16:1;0-18:2;0)	0.323	0.064	0.338	0.063
PCO(17:0;0-17:1;0)	0.296	0.058	0.297	0.058
PCO(18:0;0-14:0;0)	0.382	0.065	0.369	0.064
PCO(18:0;0-20:4;0)	0.223	0.063	0.231	0.063
PCO(18:1;0-16:0;0)	0.332	0.062	0.325	0.062
PCO(18:1;0-18:2;0)	0.306	0.064	0.318	0.062
PCO(18:1;0-20:3;0)	0.233	0.063	0.233	0.063
PCO(18:1;0-20:4;0)	0.243	0.059	0.233	0.059
PCO(18:2;0-16:0;0)	0.282	0.063	0.305	0.062
PCO(18:2;0-18:1;0)	0.335	0.072	0.357	0.070
PCO(18:2;0-18:2;0)	0.305	0.062	0.305	0.062
SM(32:1;2)	0.353	0.060	0.351	0.059
SM(34:0;2)	0.411	0.061	0.394	0.061
SM(34:1;2)	0.313	0.059	0.318	0.058
SM(34:2;2)	0.375	0.064	0.382	0.063
SM(36:1;2)	0.320	0.070	0.341	0.068
SM(36:2;2)	0.410	0.069	0.428	0.068
SM(38:1;2)	0.272	0.068	0.288	0.066
SM(38:2;2)	0.350	0.073	0.373	0.071
SM(40:1;2)	0.307	0.064	0.312	0.063
SM(40:2;2)	0.241	0.063	0.238	0.062
SM(42:2;2)	0.329	0.061	0.313	0.060
CER(40:1;2)	0.397	0.062	0.384	0.061
CER(40:2;2)	0.355	0.061	0.354	0.061
CER(42:1;2)	0.389	0.060	0.379	0.061
CER(42:2;2)	0.391	0.060	0.371	0.060
PI(16:0;0-18:1;0)	0.222	0.064	0.220	0.063
PI(16:0;0-18:2;0)	0.106	0.062	0.099	0.062
PI(18:0;0-18:1;0)	0.197	0.061	0.192	0.060
PI(18:0;0-18:2;0)	0.167	0.063	0.156	0.062
PI(18:0;0-20:3;0)	0.197	0.059	0.203	0.060
PI(18:0;0-20:4;0)	0.308	0.065	0.314	0.065
PI(18:1;0-18:1;0)	0.179	0.063	0.194	0.063
PE(18:0;0-18:2;0)	0.333	0.072	0.331	0.072
PE(18:0;0-20:4;0)	0.362	0.067	0.361	0.067
PEO(16:1;0-18:2;0)	0.344	0.064	0.344	0.064
PEO(16:1;0-20:4;0)	0.356	0.065	0.357	0.065
PEO(18:1;0-18:2;0)	0.389	0.059	0.388	0.059
PEO(18:2;0-18:2;0)	0.333	0.066	0.334	0.066
PEO(18:2;0-20:4;0)	0.337	0.065	0.338	0.065
LPE(16:0;0)	0.404	0.064	0.392	0.064
LPE(18:1;0)	0.230	0.061	0.230	0.061
LPE(18:2;0)	0.191	0.058	0.201	0.059
LPE(20:4;0)	0.266	0.063	0.262	0.063
LPE(22:6;0)	0.445	0.061	0.434	0.061
LPC(14:0;0)	0.279	0.060	0.280	0.060
LPC(16:0;0)	0.432	0.060	0.419	0.059
LPC(16:1;0)	0.259	0.059	0.260	0.059
LPC(18:0;0)	0.337	0.064	0.327	0.062
LPC(18:1;0)	0.385	0.065	0.382	0.064
LPC(18:2;0)	0.293	0.063	0.302	0.063
LPC(20:3;0)	0.307	0.067	0.307	0.067
LPC(20:4;0)	0.418	0.067	0.381	0.067
LPC(22:6;0)	0.502	0.062	0.503	0.062
ST	0.258	0.060	0.258	0.060

10. Page 6: I suggest deleting “Moreover, lipid species with genome-wide significant association had higher heritability estimates compared to the lipid species with no

significant association”, as this is evident, and at this point genome-wide significance has not been formally introduced.

We have removed this text from the manuscript as suggested.

11. The PheWeb database is a great addition to the paper – the link should be replaced by a domain name to be persistent after publication (not an IP address, which may change). Some of the links to the Oxford Big server do not work – the alleles are inverted.

We thank the reviewer for pointing this out. We have now updated the Pheweb browser after consistently aligning the alleles and all the links are working fine now. We agree that it would be nice to set up the domain for Pheweb browser. We are working on the technicality of it and will soon replace the current IP address (<http://35.205.141.92>) with the domain name.

12. The association of the ROCK1 variant with CVD appears stretched if there is no genetic signal in large CVD GWAS. The association of BLK with diabetes also appears quite stretched ($P=4.5E-5$) and should be substantiated ... the same holds for GALNT16 and LPL ... in this context would it be interesting to be more formal about how many tests with disease have been made.

In the PheWAS analyses, 25 CVD phenotypes and 34 lead variants (one lead variant was not available) were included. As the CVD phenotypes are correlated, we applied false discovery rate (FDR) for multiple testing correction as Bonferroni correction would be too conservative in this case. So, PheWAS associations with false discovery rate (FDR) <5% evaluated using the Benjamini-Hochberg method were considered significant.

As suggested, we have modified the text in the manuscript as:

Page 6, Lines 23-25: *“Among the new loci, the strongest association was at an intronic variant rs151223356 near ROCK1 with short acyl-chain LPC(14:0,0) ($P=1.9 \times 10^{-10}$). ROCK1 encodes for a serine/threonine kinase that plays key role in glucose metabolism.”*

Page 8, Lines 18-19: *“BLK (rs1478898-A) was also found to be associated with decreased risk of obesity ($OR=0.97$, $P=5.6 \times 10^{-8}$) and type 2 diabetes ($OR=0.96$, $P=4.5 \times 10^{-5}$).”*

13. Figure 2D: Delete Y-axis legend “Height”

Thank you for the suggestion. We have taken care of this in the revised manuscript.

14. Figure 4B: X-axis legend is too small

We have taken care of the font sizes in all the figures in the revised manuscript.

Reviewer #2

The authors report results of the hitherto largest genome-wide study of lipidomic profiles. More specifically, they studied 141 lipid species, followed by a phenome-wide scan with 26 CVD related phenotypes. They identified 35 loci, 15 of them not reported with lipid-related phenotypes, yet. 10 of the loci were also related with CVD endpoints. Furthermore, they also showed highly heterogeneous estimates of heritability and variance explained.

Altogether, this is a very interesting, timely manuscript, which has been very thoroughly performed with state-of the art methods.

I only have one major point:

We thank the reviewer for the kind words and valuable suggestions.

1. With individual GWAS on 141 lipid species and 4 traditional lipid measures, defining a p-value of smaller than 5×10^{-5} as genome-wide significant is not really appropriate. Of course, the correlation between the traits has to be taken into account, e.g. by correcting for principal components explaining the majority of the correlation between traits, as has been done in recently published metabolomics GWAS.

As suggested, we have now adjusted the p-values for the number of principal components (PC) that explain over 90% of the variance in the lipidomics data. After accounting for 34 PCs that explain 90% of the variance, we set the p-value threshold to be smaller than 1.5×10^{-9} .

The method section now includes the following:

Page 20, Lines 3-5: *“To account for multiple tests, the study-wide P value threshold was set at $<1.5 \times 10^{-9}$ after correcting for 34 principal components (PCs) that explain over 90% of the total variances in lipidomic profiles.”*

And the association results have been modified based on this p-value cut off (Pages 6 and 7)

2. Some additional minor points: Please check the numbering of the Supplementary Tables in the text, e.g. on page 20, it should probably be Suppl. Table 6 instead of 4, page 5: Suppl. Table 2 instead of 1.

Thanks for pointing this out. We have taken care of this in the revised manuscript.

3. Figure 3: what is the rationale of ordering?

In Figure 3, the lipid species are ordered based on the hierarchical clustering using genetic and phenotypic correlations (lower panel of the Figure 3) and their heritability estimates and proportion of variance explained for the lipid species is shown in upper panel in the same order.

We have clarified this in the legend of the figure 3 in the revised manuscript.

Page 30, Lines 22-23: *“The lipid species are ordered based on the hierarchical clustering showing the correlations between the lipid species and traditional lipids.”*

4. All Figures and supplementary figures: Make sure they are readable in the final submitted version (maybe enlarge axis labels, if possible), e.g. not readable at the moment: y-axis in Figure 4 B, figure 6 A and B, suppl. Figure 5 and 6

Thanks for pointing this. We have taken care of the font sizes in all the figures in the revised manuscript and hope that they are readable in the current form.

REVIEWERS' COMMENTS:

Reviewer #1 (Remarks to the Author):

The authors did a really great job in revising the manuscript. From my point of view it is ready for publication.

Reviewer #2 (Remarks to the Author):

I am happy with the changes the authors made. I have no more comments.